# The deep-rooted origin of disulfide-rich spider venom toxins

Naeem Yusuf Shaikh, Kartik Sunagar*

Evolutionary Venomics Lab, Centre for Ecological Sciences, Indian Institute of Science Bangalore, Bengaluru, India

**Abstract** Spider venoms are a complex concoction of enzymes, polyamines, inorganic salts, and disulfide-rich peptides (DRPs). Although DRPs are widely distributed and abundant, their bevolutionary origin has remained elusive. This knowledge gap stems from the extensive molecular divergence of DRPs and a lack of sequence and structural data from diverse lineages. By evaluating DRPs under a comprehensive phylogenetic, structural and evolutionary framework, we have not only identified 78 novel spider toxin superfamilies but also provided the first evidence for their common origin. We trace the origin of these toxin superfamilies to a primordial knot – which we name 'Adi Shakti', after the creator of the Universe according to Hindu mythology – 375 MYA in the common ancestor of Araneomorphae and Mygalomorphae. As the lineages under evaluation constitute nearly 60% of extant spiders, our findings provide fascinating insights into the early evolution and diversification of the spider venom arsenal. Reliance on a single molecular toxin scaffold by nearly all spiders is in complete contrast to most other venomous animals that have recruited into their venoms diverse toxins with independent origins. By comparatively evaluating the molecular evolutionary histories of araneomorph and mygalomorph spider venom toxins, we highlight their contrasting evolutionary diversification rates. Our results also suggest that venom deployment (e.g. prey capture or self-defense) influences evolutionary diversification of DRP toxin superfamilies.

## Editor's evaluation

This is an important survey of disulfide-rich peptides (DRPs), which comprise a large fraction of the most functionally important components of spider venom. While spider DRPs were thought to have evolved independently numerous times throughout the spider tree of life, the authors make a solid case for the idea that they all stem from a single common ancestral protein. The study makes a significant advance towards formalizing the diversity of spider venoms, which will be of interest both to scientists working on protein evolution and to those working on functional venomics.

*For correspondence: ksunagar@iisc.ac.in

## Introduction

With their killer instinct and deadly toxins, spiders have been at the centre of many myths and folktales from times immemorial. They are an archetypal arthropod group with mid-Cambrian or early Ordovician origin, nearly 495 million years ago (MYA; **Lozano-Fernandez et al., 2016**). Because of their unique ability to secrete silk and venom, spiders have successfully colonised diverse ecological niches. They are amongst the most successful predators on the planet, with over 50,000 species and 129 families described to date (**King, 2004**; **WSC, 2022**). The majority of spiders are equipped with chelicerae harbouring venom glands, with Symphytognathidae, Uloboridae, and certain Mesothelae species being the only exceptions (**King, 2004**; **Mullen and Vetter, 2019**).

Spider venoms are a concoction of enzymes, polyamines, nucleic acids, inorganic salts and disulfiderich peptides (DRPs) (**Senji Laxme et al., 2019b**; **King and Hardy, 2013**). They are predominantly rich

**eLife digest** The majority of spiders rely on their venom to defend themselves, to hunt, or both. Armed with this formidable weapon, they have managed to conquer every continent besides Antarctica since they first emerged about 495 million years ago.

A closer look at spider venoms hints at an intriguing evolutionary history which has been rarely examined so far. The venom of other animals, such as snakes or scorpions, is usually formed of a wide range of unrelated toxins; in contrast, spiders rely on a single class of proteins, known as disulfide-rich peptides, to create their deadly venom cocktail. This family of molecules is impressively diverse, with each peptide having a distinct structure and mode of action. Its origins, however, have remained elusive.

To fill this knowledge gap, Shaikh and Sunagar scanned the sequences of all disulfide-rich peptides generated to date, bringing together a dataset that includes 60% of all modern-day spiders. The analyses allowed the identification of 78 new superfamilies of spider toxins. They also revealed that all existing peptides originate from a single molecule, which Shaikh and Sunagar named after the powerful Hindu goddess Adi Shakti. This ancestral toxin was present 375 million years ago in the last common ancestor of modern-day spiders.

The work also highlighted that disulfide-rich peptides evolved under different pressures in various groups of spiders; this may be because some species primarily use their venom for hunting, and others for defence. While the 'hunters' may need to constantly acquire toxins with new roles and structures to keep their edge over their prey, those that rely on venom to protect themselves may instead benefit from relying on tried-and-tested toxins useful against a range of infrequent predators. Finally, the analyses revealed that the disulphide-rich peptides of Mygalomorphae tarantulas, which form one of the three major groups of spiders, are much more diverse than the related toxins in other spiders. The underlying reason for this difference is still unclear.

Several life-saving drugs currently on the market are based on toxins first identified in the venoms of snakes, cone sails or lizards. Similar discoveries could be unlocked by better understanding the range of deadly molecules used by spiders, and how these came to be.

in DRPs that are characterised by a diversity of structural motifs, including Kunitz (*Yuan et al., 2008*), disulfide-directed β-hairpin (*Wang et al., 2000*), disulfide-stabilised antiparallel β-hairpin stack (DABS; *Pineda et al., 2020*) and inhibitor cystine knot (ICK) – also known as knottins (*Pallaghy et al., 1994*; *Undheim et al., 2016*). Despite the fact that DRPs constitute three-quarters of spider venom, our evolutionary understanding of their origin and diversification has remained elusive. This knowledge gap stems from a lack of sequence and structural data for DRPs from diverse spider lineages and the prevalence of significant sequence divergence in these toxins.

Here, we examined DRP sequences from the spiders of the Mygalomorphae infraorder (includes funnel-web spiders and tarantulas), family Theridiidae (includes the red-back spiders), and the Retro-lateral Tibial Apophysis (RTA) clade from Araneomorphae, which constitutes over 58% of spider genera (2527 genera) described to date (*Figure 1A*). A molecular phylogenetic framework implemented in this study resulted in the identification of 78 novel toxin superfamilies and suggests a deep-rooted origin of venom DRPs in spiders. Our findings also highlight the role of distinct prey capture strategies of Araneomorphae and Mygalomorphae in shaping the recruitment and diversification of venom DRPs. Furthermore, by comparatively evaluating spider venom toxins employed for anti-predatory defense and prey capture, we also unravel the impact of the purpose of venom deployment on the evolution of spider venoms. Thus, sequence, phylogenetic, structural and evolutionary assessments in this study have provided insights into the fascinating origin and early diversification of this predominant spider venom component.

## Results
### Novel spider toxin superfamilies

Superfamilies (SF) of venom toxins in spiders have been classified based on their signal peptide and propeptide sequences (*Pineda et al., 2014*). This premise was first used to describe the Shiva

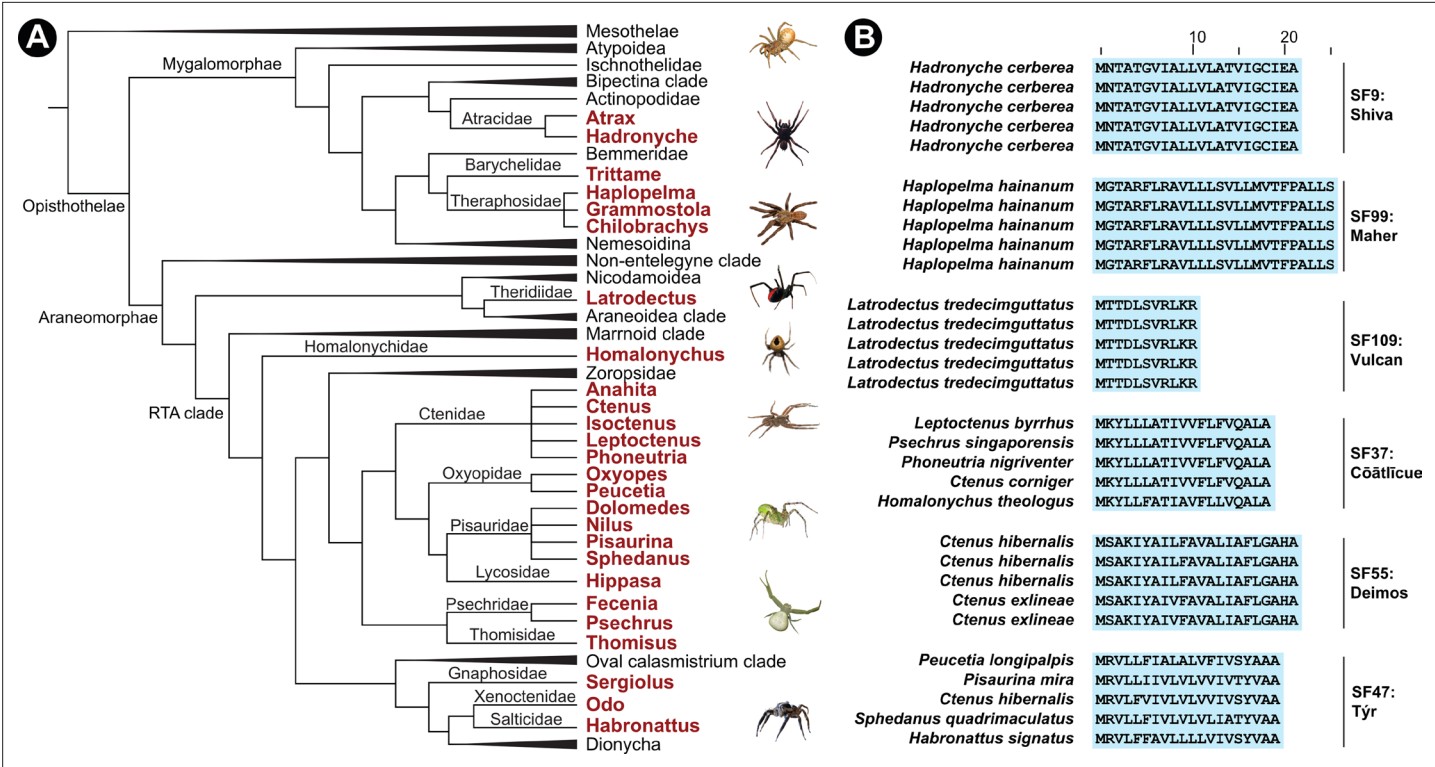

**Figure 1.** Schematic representation of Araneae phylogeny and their venom superfamilies. Panel A here shows a cladogram of Araneae with lineages under investigation indicated in red. In panel B, representative signal peptide alignments of toxin superfamilies are shown with sequence conservation of >90% highlighted in blue.

superfamily of toxins from Atracidae spiders (*Pineda et al., 2014*). Recently, using a similar approach, 33 novel spider toxin superfamilies have been identified from the venom of the Australian funnel-web spider, *Hadronyche infensa* (*Pineda et al., 2020*). Since gene phylogenies have not been extensively utilised while classifying spider venom toxins, our understanding of their origin and diversification has been severely limited.

In this study, we relied on the strong conservation of signal peptide and propeptide regions in identifying several novel spider venom toxin superfamilies, following the same strategy as [*Figure 1B*; *Pineda et al., 2020*; *Pineda et al., 2014*]. Blast searches were used to identify the homology between largely divergent toxin superfamilies. Toxin sequences were found to share strong sequence conservation within a superfamily. Cysteine residues, which are involved in the formation of disulphide bonds and, thereby, are extremely vital in determining protein structure and function, were used as guides to manually refine sequence alignments. This approach enabled the identification of 33 novel toxin superfamilies along the breadth of Mygalomorphae (*Figure 2—figure supplements 2 and 3*). Among these, 31 superfamilies belonged to the DRP class, whereas the other two were enzymatic non-DRP toxins, including the first report of Neprilysin (SF103) and CAP (CRiSP/Allergen/PR-1; SF104) from Atracidae spiders (*Figure 4—source data 1*).

Moreover, analyses of Araneomorphae toxin sequences using the strategy above resulted in the identification of 45 novel toxin superfamilies from Araneomorphae, all of which but one (SF109) belonged to the DRP class of toxins (*Figure 3—figure supplements 2 and 3*). Overall, among all novel spider toxin superfamilies identified in this study, the majority (n=75) were DRPs, reinstating the dominance of this toxin type in spider venoms. Based on the arrangement of cysteine residues involved in the formation of disulphide bonds, these DRPs could be further segregated into ICK-like (n=28), DABS (n=13) and novel disulphide patterned non-ICK (n=34) superfamilies (*Pineda et al., 2020*). We named these novel spider toxin superfamilies after deities of death, destruction, and the underworld (*Supplementary file 1*), following a nomenclature system introduced by *Pineda et al., 2014*.

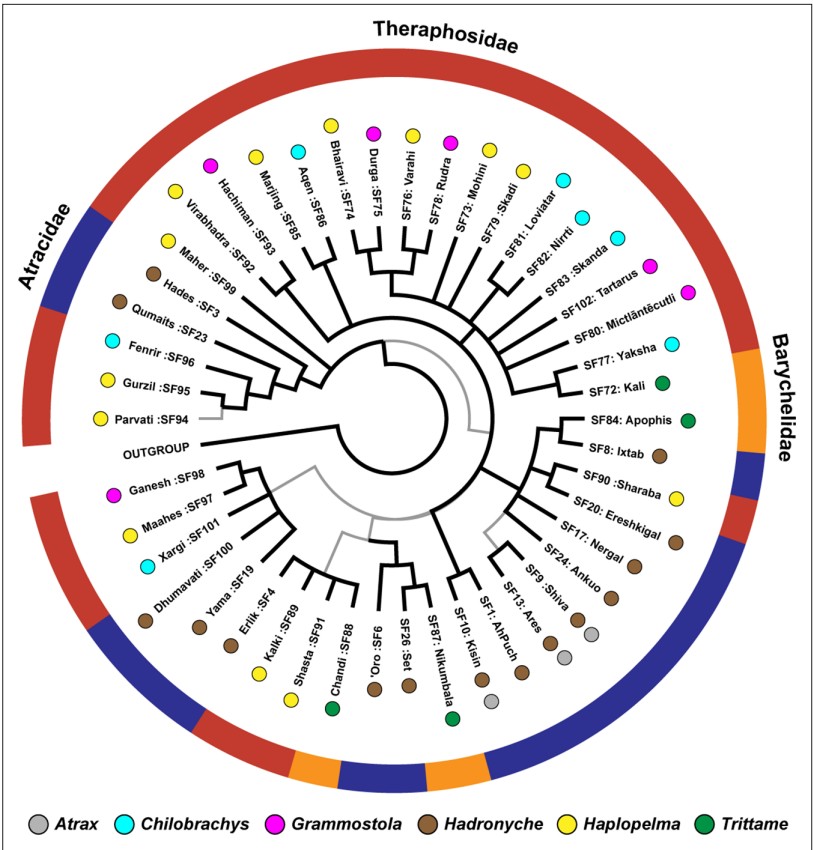

**Figure 2.** The Bayesian phylogeny of mygalomorph spider venom toxin superfamilies. This figure represents the Bayesian phylogeny of Mygalomorphae spider toxin superfamilies, where branches with high (BPP >0.95) and low (BPP <0.95) node supports are shown in thick black and thin grey lines, respectively. Coloured spheres alongside tree tips represent the spider genera, while the coloured outer circle indicates the spider family in which the respective toxin superfamily has been identified (Atracidae [red], Barychelidae [orange], and Theraphosidae [blue]).

The online version of this article includes the following source data and figure supplement(s) for figure 2:

**Source data 1.** This zip archive contains sequence alignment used to perform phylogenetic analyses and the Bayesian phylogeny of Mygalomorphae DRP toxin superfamilies.

**Figure supplement 1.** Phylogeny of Mygalomorphae spider toxin superfamilies.

**Figure supplement 2.** Signal peptide and propeptide alignment of novel mygalomorph superfamilies.

**Figure supplement 3.** Homology models of novel Mygalomorphae toxin superfamilies.

**Figure supplement 4.** Principal component analyses for Mygalomorphae toxin superfamilies.

The identification of novel toxin superfamilies was further supported by phylogenetic and principal component analyses. Reconstruction of evolutionary histories using Bayesian inference (BI) and maximum-likelihood (ML) approaches retrieved monophyletic groups of toxin superfamilies (*Figures 2 and 3*; node support: ML:>90/100; BI:>0.95; refer to figure supplement provided for complete phylogeny with branch lengths). Interestingly, the plesiotypic DRP scaffold seems to have undergone lineage-specific diversification in Mygalomorphae, where the selective diversification of the scaffold has led to the origination of novel toxin superfamilies corresponding to each Linnaean genus (*Figure 2*). In our Bayesian and maximum-likelihood phylogenetic tree reconstructions, these toxin scaffolds were found to form distinct clades, further supporting this claim (*Figure 2—figure supplement 1*; node support: ML: ML:>90/100; BI:>0.95).

A similar pattern was also observed in the case of Araneomorphae, where certain toxin SFs (n=12) were found to have diversified within individual genera, corresponding to the Linnaean taxonomy (*Figure 3*). However, we also documented a large number of DRP toxins (n=32) that were found to have diversified in a family-specific manner, wherein, a toxin scaffold seems to have a more ancient

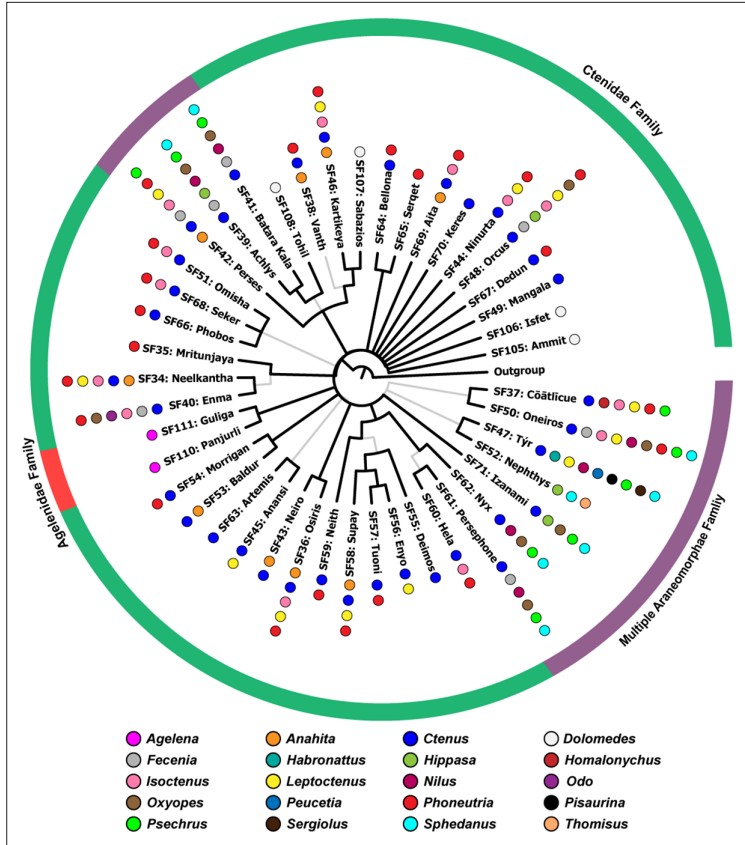

**Figure 3.** The Bayesian phylogeny of araneomorph spider venom toxin superfamilies. This figure represents the Bayesian phylogeny of Araneomorphae spider toxin superfamilies, where branches with high (BPP >0.95) and low (BPP <0.95) node supports are shown in thick black and thin grey lines, respectively. Coloured spheres, alongside tree tips, represent the spider genera, while the coloured outer circle indicates the spider family (Agelenidae [red], Ctenidae [green], multiple araneomorph families [purple]: Ctenidae, Gnaphosidae, Homalonychidae, Lycosidae, Oxyopidae, Pisauridae, Psechridae, Salticidae, Thomisidae, Xenoctenidae) in which the respective toxin superfamily has been identified.

The online version of this article includes the following source data and figure supplement(s) for figure 3:

**Source data 1.** This zip archive contains sequence alignment used to perform phylogenetic analyses and the Bayesian phylogeny of Araneomorphae DRP toxin superfamilies.

**Figure supplement 1.** Phylogeny of Araneomorphae spider toxin superfamilies.

**Figure supplement 2.** Signal peptide alignment of araneomorph superfamilies.

**Figure supplement 3.** Homology models of novel Araneomorphae toxin superfamilies.

**Figure supplement 4.** Principal component analyses for Araneomorphae toxin superfamily.

recruitment, corresponding to the Linnean family, rather than the Linnaean genus. As a result, and in contrast to mygalomorph DRPs, araneomorph toxin superfamilies were found to be scattered across spider lineages (*Figure 3*; *Figure 3—figure supplement 1*; node support: ML:>90/100; BI:>0.95). Moreover, Principal component analysis (PCA) of toxin sequences further provided evidence for the monophyly of mygalomorph and araneomorph SFs, where each toxin superfamily formed a distinct group in PCA plots (*Figure 2—figure supplement 4*; *Figure 3—figure supplement 4*).

Furthermore, sequence alignments of DRPs clearly highlighted the homology among DRP toxin superfamilies (*Figure 4*; *Figure 4—figure supplement 1*; node support: ML:>90/100; BI:>0.95). Six cysteine residues were found to be nearly universally conserved across 101 DRP toxin SFs (*Figure 4B*; *Figure 4—figure supplement 2*). Our findings enabled us to trace the origin of spider venom DRPs in Opisthothelae, the clade that encompasses Araneomorphae and Mygalomorphae (*Magalhaes et al., 2020*). Thus, we highlight for the first time that all DRP toxins in spiders may have had a common

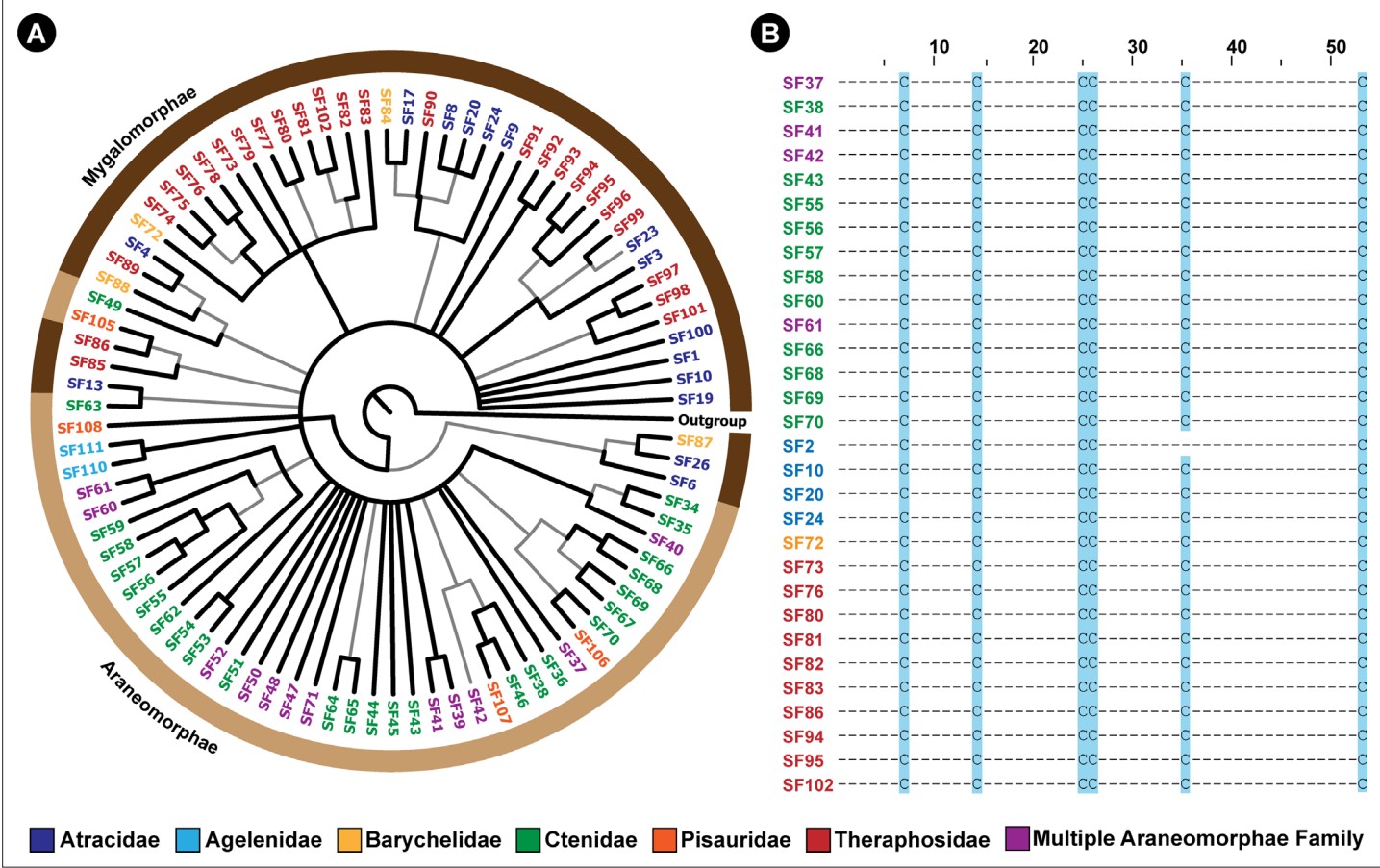

**Figure 4.** The Bayesian phylogeny and cysteine framework representation of spider venom DRPs. This figure depicts the Bayesian phylogeny and alignment of representative sequences of Araneae DRP toxin superfamilies, where branches with high (BPP >0.95) and low (BPP <0.95) node supports are shown in thick black and thin grey lines, respectively. The coloured outer circle in panel A indicates the infraorder of spiders (Mygalomorphae and Araneomorphae shown in dark and light brown, respectively) in which the respective DRP superfamily was identified. In panel B, cysteine framework conserved across toxin SFs is highlighted in blue.

The online version of this article includes the following source data and figure supplement(s) for figure 4:

**Source data 1.** This zip archive contains sequence alignment used to perform phylogenetic analyses and the Bayesian phylogeny of Araneae DRP toxin superfamilies.

**Figure supplement 1.** Phylogeny of Araneae spider toxin superfamilies.

**Figure supplement 2.** Mature peptide alignment of mygalomorph and araneomorph DRP superfamilies.

molecular origin, nearly 375 MYA. It should be noted, however, that functional analyses have been performed only on a handful of mygalomorph toxins, with even fewer studies focusing on araneo-morph toxin superfamilies, and that it would be inaccurate to speculate on the functions of these toxins based on homology.

## Molecular evolution of spider venom DRP toxins

To evaluate the nature and strength of the selection that has shaped spider venom DRPs, we employed site-specific models that detect selection across sites. Our findings suggest that the majority of Myga-lomorphae toxin superfamilies (12/19 SFs) have evolved under the influence of positive selection ($\omega$ ranging between 1.1 and 2.9; positively selected sites [PS]: 0–26), while the remaining few have experienced negative or purifying selection ($\omega$ ranging between 0.7 and 0.8; PS: 0–13; *Figure 5*, *Figure 5—source data 1*). In stark contrast, nearly all of the Araneomorph toxin superfamilies that we investigated here were found to have evolved under a strong influence of negative selection ($\omega$ ranging between 0.2 and 1.0; PS: 0–10; *Figure 5*, *Figure 5—source data 1*). We further assessed whether these changes documented across sites have a significant effect on the biochemical and

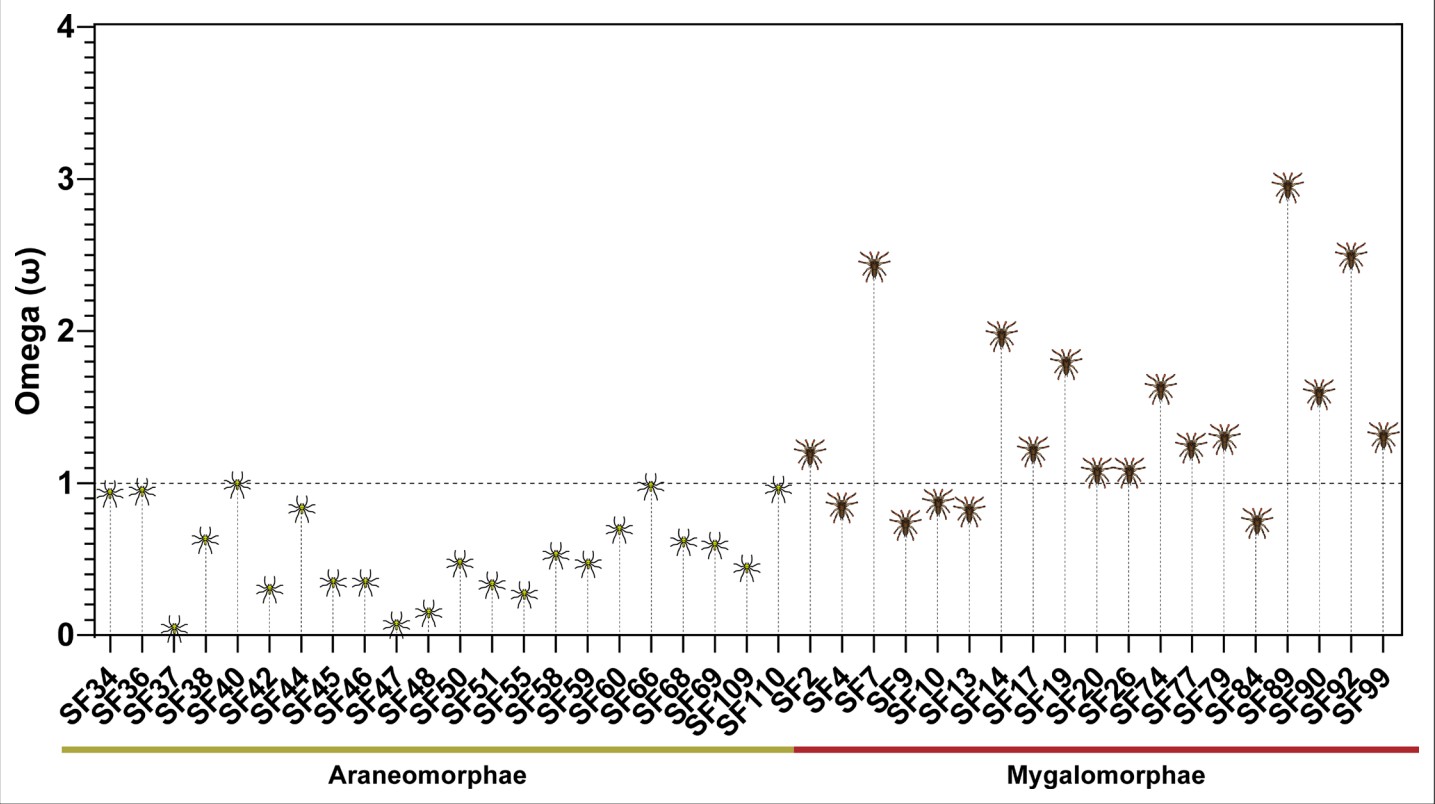

**Figure 5.** Molecular evolution of spider toxin superfamilies. This figure shows the distribution of $\omega$ values (Y-axis) for araneomorph and mygalomorph spider venom toxin superfamilies (X-axis). The horizontal dotted black line represents neutral evolution ($\omega$=1), with $\omega$ values above and below it indicating positive ($\omega$>1) and negative ($\omega$<1) selection, respectively.

The online version of this article includes the following source data and figure supplement(s) for figure 5:

**Source data 1.** Molecular evolution of toxin superfamilies.

**Figure supplement 1.** Deployment strategies dictate the evolution of spider venom.

structural properties of amino acids using TreeSAAP (*Figure 5—source data 1*). Outcomes of these analyses revealed the accumulation of replacement changes in Mygalomorphae toxin superfamilies that result in radical shifts in amino acid properties, potentially influencing their structure and function (*Figure 5—source data 1*).

To comparatively evaluate the nature of selection that shapes venom components deployed either for prey capture or antipredator defence, we employed maximum-likelihood and Bayesian approaches. In these analyses, we identified toxin superfamilies SF74, SF77, SF79, SF89, SF90, SF92, and SF99 as predatory toxins (i.e. toxins deployed for prey capture – refer to the discussion section for the principle considered for this classification), whereas SF13 (i.e. Ares SF) was classified as a defensive spider venom toxin superfamily (i.e. toxins deployed for antipredator defence) as described previously (*Herzig et al., 2020*). Assessment of molecular evolutionary regimes identified a significant influence of positive selection on venom toxins that are employed for prey capture ($\omega$ ranging between 1.2 and 2.9; PS: 0–11, *Figure 5—source data 1*, *Figure 5—figure supplement 1*), relative to those that are chiefly or exclusively used for antipredatory defence ($\omega$=0.8; PS: 3; *Figure 5—source data 1*, *Figure 5—figure supplement 1*).

## Discussion
### The deep evolutionary origin and diversification of the primordial knot
Prior attempts to explore the phylogenetic and evolutionary histories of spider venom DRPs have hypothesised independent origin and lineage-specific diversification of DRP venom toxins (*Rodríguez de la Vega, 2005*). In contrast, recent literature, primarily focusing on *Hadronyche infensa*, suggests

that the diverse disulfide-rich venom arsenal of this Australian funnel-web spider is a derivative of an ancestral ICK motif that underwent several rounds of duplication and diversification (*Pineda et al., 2020*). Often restricted to a specific spider lineage, or given the inconsistent ways of classifying spider venom toxins, previous attempts have failed to provide a broader perspective on the evolution of these peptides (*Ferrat and Darbon, 2005*; *Chen et al., 2008*). Given their very long evolutionary histories, genes encoding DRP toxins have undergone significant diversification, making it difficult to precisely trace their phylogenies. Together with the lack of structural and functional data for these toxins, all of the aforementioned factors have impeded our understanding of the origin and evolution of this predominant spider venom component.

To address this knowledge gap, we employed sequence comparisons, phylogenetic inferences and evolutionary analyses, which enabled the identification of 76 novel spider venom toxin superfamilies (45 from Araneomorphae [44 DRP and 1 non-DRP] and 33 from Mygalomorphae spiders [31 DRP and 2 non-DRP]). Our findings strongly suggest a deep-rooted origin of DRP spider venom toxin superfamilies (*Figure 4A*), possibly from a single ancestral DRP or knottin scaffold, which we name 'Adi Shakti', after the original creator of the universe according to Hindu mythology. We propose that all of the extant spider disulfide-rich peptide toxin superfamilies (n=101) in Mygalomorphae and Araneomorphae, which include those that were previously reported (n=26), as well as the ones identified in the present study (n=75), have originated from this 'primordial knot', further undergoing lineage-specific gene duplication and diversification (*Figures 2–4*). The origin and diversification of these superfamilies can be explained by a mechanism that is similar to the combinatorial peptide strategy, wherein certain venomous animals, such as cone snails, generate a remarkable diversity in their mature toxin peptides while preserving the signal and propeptide regions (*Escoubas and Rash, 2004*; *Zhu et al., 2000*; *Olivera et al., 1995*). Rapid events of diversification, preceded by repeated rounds of gene duplication, form the basis of the combinatorial peptide library strategy (*Sollod et al., 2005*). These hypermutational events have been previously shown to be restricted to the mature peptide regions of toxins (*Conticello et al., 2001*). In contrast, the signal and propeptide regions, which are vital for the precise secretion and folding of proteins, respectively, evolve under the strong influence of negative selection pressures (*Duda and Palumbi, 1999*) - a molecular evolutionary trend also reported in venom coding genes of snakes (*Brust et al., 2013*). Spider venom coding genes appear to have followed a similar strategy. However, unlike the cone snail venom coding genes that have a recent evolutionary origin (<35–50 MYA; *Olivera, 1997*; *Duda and Kohn, 2005*), spider venom toxins have likely originated from an ancestral scaffold in Opisthothelae, the clade that encompasses Mygalomorphae and Araneomorphae spiders (over 99% of all spider genera described to date), nearly 375 MYA (*Magalhaes et al., 2020*). Given their significant sequence divergence since their deep-rooted evolutionary origin, the entire protein-coding gene, including the signal and propeptide regions, has accumulated significant differences. Consistent with this hypothesis, the majority of positively selected sites (~96%) identified in spider venom DRP toxins (all sites in Araneomorphae, and all but two sites in Mygalomorphae) were restricted to the mature peptide region, whereas the signal and propeptide regions harboured a minor proportion of these sites (1% and 3%, respectively; *Figure 5—source data 1*).

It has been theorised that the plesiotypic (or ancestral) DRP scaffold comprised of eight cysteines that formed four disulfide bonds (*Pineda et al., 2020*; *Cole and Brewer, 2021*). However, the evolutionary history of DRP scaffold in spider venom has been riddled with events of duplication and diversification, which may have resulted in multiple gain and loss of structural and functional residues, including cysteines. As a result, we find that DRP toxins in extant spiders are comprised of distinct scaffolds with a range of cysteine pairs (4–12 cysteines forming 2–6 disulfide bonds). For example, an individual spider from the *Haplopelma* genus may contain SF90, SF91, SF99 and SF100 toxin superfamilies in its venom with 6, 8, 12, and 10 cysteines, respectively. This makes it very difficult to trace the nature and cysteine skeletal structure of the plesiotypic scaffold – something that could be answered in future with the help of comparative genomics and synteny analyses. Our extensive phylogenetic and evolutionary analyses provide insight into the common origin of DRP toxins in spiders, dating back to the common ancestor of Mygalomorphae and Araneomorphae. However, we refrain from speculating on the exact nature of this plesiotypic scaffold.

## Contrasting weaponisation strategies: recruitment versus innovation

Venom is an intrinsically ecological trait that has underpinned the evolutionary success of many animals (*Suranse et al., 2022*). The ability of venomous organisms to incapacitate prey and predators

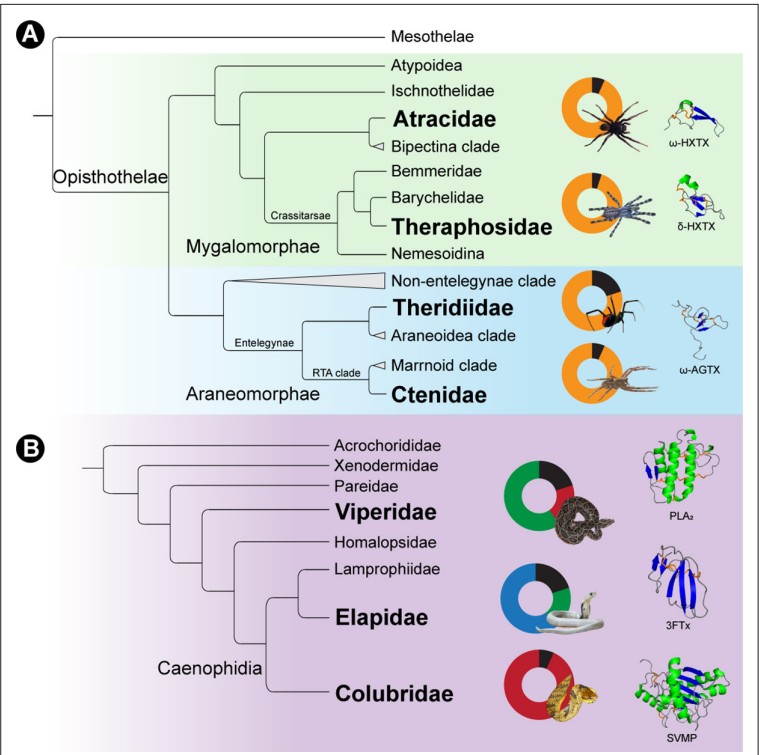

**Figure 6.** Distinct toxin scaffold recruitment strategies in spiders and snakes. This figure depicts distinct toxin scaffold recruitment strategies in (**A**) spiders and (**B**) advanced snakes. The Araneae phylogeny highlights the domination of disulfide-rich peptide toxins in spiders [Atracidae: *Atrax sp.*; Theraphosidae: *Poecilotheria formosa*; Theridiidae: *Latrodectus mactans*; Ctenidae: *Phoneutria nigriventer*: e.g., **Palagi et al., 2013**; **Oldrati et al., 2017**; **Diniz et al., 2018**], whereas venoms of advanced snakes are constituted by diverse phylogenetically unrelated toxin superfamilies (Viperidae: *Daboia russelii*, Elapidae: *Naja naja*, Colubridae: *Spilotes sulphureus*: e.g., **Senji Laxme et al., 2021a**; **Senji Laxme et al., 2021b**; **Modahl et al., 2018**). Doughnut charts, portraying the major molecular scaffolds in venom are also shown disulfide-rich peptides (yellow), snake venom metalloproteinases (SVMP, red), phospholipase $A_2$ (PLA$_2$, green), three-finger toxins (3 FTx, blue) and other minor components (black). Structures of the major scaffolds are also shown, with helices coloured in green, β-strands in blue and disulfide bonds in orange.

emanates from toxins that exhibit an array of biochemical activities and target divergent pathways. Many venomous lineages deploy a wide range of toxins from phylogenetically unrelated superfamilies. Venomous snakes, for example, have 'recruited' a myriad of toxins, including snake venom metalloproteinases, snake venom serine proteases, three-finger toxins, phospholipase $A_2$s, L-amino acid oxidases, Kunitz-type serine protease inhibitors, kallikreins, lectins, DNases, and hyaluronidases (**Casewell et al., 2013**; **Faisal et al., 2021**; **Dutta et al., 2017**; **Senji Laxme et al., 2019a**; **Casewell et al., 2020**; **Figure 6**). Similarly, spider venoms typically possess many forms of enzymes (e.g. phospholipases, proteases and chitinases), polyamines, salts, and disulphide-rich toxins (**King and Hardy, 2013**; **Figure 6**). However, spider venom DRPs with diverse ion channel targeting activities, such as sodium, potassium, calcium, and chloride ion channels, predominate in the venoms of nearly all spiders, constituting three-quarters of the venom (**Figure 6**). Phylogenetic and evolutionary assessments in this study trace the evolutionary origin of DRPs in Opisthothelae, the suborder that includes the majority of spiders described to date. This strategy, wherein, a molecular scaffold with a single deep-rooted evolutionary origin constitutes the major content of the venom, is unique to spiders. Venoms of most other animals are, instead, composed of unrelated toxin types, derived from distinct scaffolds/gene superfamilies in varying proportions. Thus, instead of recruiting distinct toxins with diverse functions into their venoms like the majority of venomous animals, spiders seem to have diversified a single molecular template to generate a commensurate functional diversity in their venoms. These findings not only shed light on the fascinating evolutionary history of spider venoms but also

highlight an unrealized potential of molecular scaffolds in underpinning the dramatic structural and functional diversification of the venom arsenal.

## Distinct recruitment and diversification of spider venom superfamilies in Mygalomorphae and Araneomorphae

In addition to suggesting the common evolutionary origin of DRP toxins, Bayesian and maximum-likelihood phylogenies provided intriguing insights into the early diversification of DRPs in spiders. Mygalomorph DRP toxin superfamilies formed lineage-specific toxin clades (63/66) that suggested the recruitment of unique DRP scaffolds at the level of Linnaean genera (*Figure 2*), while the majority of unique DRP scaffolds seemed to be recruited ancestrally at the level of Linnaean families in Araneomorphae (*Figure 3*). Only a minor fraction (6/38) of araneomorph toxin superfamilies were recruited at the level of Linnaean genera.

When the nature and strength of selection on venom DRPs were assessed, a strong influence of positive selection was identified on the evolution of these toxin superfamilies in mygalomorph spiders. Only a minority of these toxin superfamilies were found to be evolving under negative selection (6/19), or under near neutral evolution (1/19), while the majority (12/19) experienced diversifying selection ($\omega$ between 1.19 and 2.95; PS: 0–26, *Figure 5*). In complete contrast, the evolution of venom superfamilies (21/22) in Araneomorphae was constrained by purifying selection ($\omega$ between 0.03 and 0.97; PS: 1–3, *Figure 5*), and a single superfamily was found to be evolving nearly neutrally ($\omega$ of 1.0; PS: 10). We further investigated the impact of these amino acid replacements on the structure and function of spider venom toxins. Outcomes of these evaluations suggest that the majority of replacements in mygalomorph spiders (between 0 and 29 properties) had a radical effect on the structure and/or biochemical property of the encoded toxin, while none were identified in most toxin superfamilies of Araneomorphae. Only a minor proportion of non-synonymous substitutions in two toxin superfamilies (SF40 and SF68) of araneomorph lineage were reported to be radically different (*Figure 5—source data 1*). Differences in the evolutionary histories of mygalomorph and the araneomorph DRP toxin superfamilies became apparent as we further evaluated them for the signatures of episodic diversification. We detected a greater prevalence of episodic diversifying selection on mygalomorph DRP toxin superfamilies than their araneomorph counterparts (0–34 versus 0–6 events, respectively).

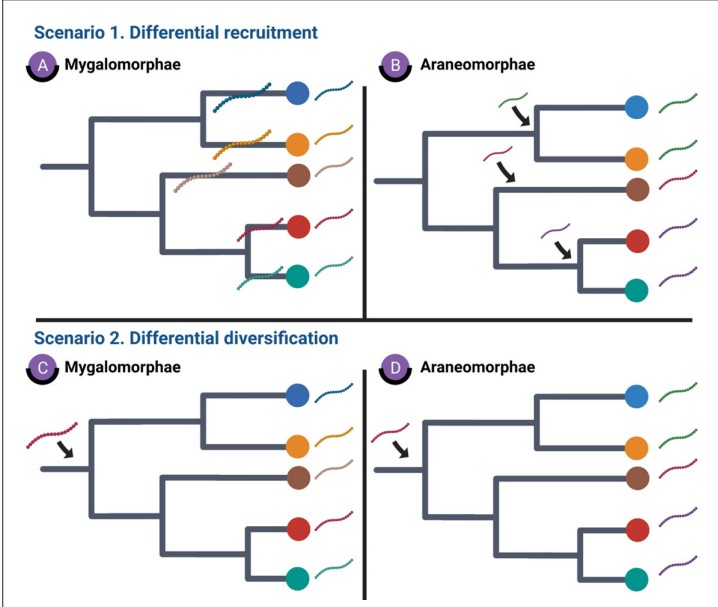

**Figure 7.** Hypotheses explaining the stark differences in recruitment and diversification of toxin SFs in Araneomorphae and Mygalomorphae. This figure depicts various hypotheses that explain distinct toxin SF recruitment and diversification in spiders. Scenario 1 depicts genus- or family-specific recruitment of spider toxin SFs in Mygalomorphae and Araneomorphae, respectively, while scenario 2 highlights the implications of differential rates of diversification.

Such starkly contrasting phylogenetic and evolutionary patterns are indicative of differential recruitment and diversification of DRPs in spiders. We postulate the following hypotheses that could possibly explain this unique pattern of spider venom evolution.

## Scenario 1. Distinct recruitment hypothesis

Mygalomorphae spiders may have recruited individual toxin superfamilies or unique toxin scaffolds post the emergence of spider family members corresponding to the Linnaean taxonomy (*Figure 7A*). This could explain why toxin superfamilies in Mygalomorphae form lineage-specific clades that correspond to individual spider genera (Linnaean taxonomy) in our phylogenetic analyses (*Figure 2*). In contrast, Araneomorphae may have recruited unique toxin scaffolds prior to the divergence of family members, insinuating an ancestral recruitment event (*Figure 7B*). This, perhaps, explains why araneomorph toxin SFs are scattered across spider families and form Linnaean family-specific groups in phylogenetic trees (*Figure 3*).

## Scenario 2. Differential molecular diversification rate hypothesis

The apparent Linnaean taxa-specific (genus- and family-) diversification of spider venom toxin superfamilies can also be explained by the differential rate of molecular evolution in the spider infraorders. In this schema, the recruitment of toxin SFs could have happened in the common ancestor of Mygalomorphae and Araneomorphae. However, the contrasting rates of diversification, wherein, mygalomorph toxin SFs underwent extensive diversification under positive selection, while araneomorph toxin SFs were very well conserved under negative selection, resulted in the contrasting patterns of DRP diversification that we see today (*Figures 5, 7C and D*).

## Scenario 3. Prey-capture strategies and toxin recruitment hypothesis

The use of webs for prey capture in araneomorphs versus the sit-and-wait predation strategy of mygalomorphs may have resulted in the selective diversification of toxin superfamilies. Since most araneomorph spiders heavily rely on their foraging web for prey capture, and because these spiders mostly prey on insects (*Pérez-Miles and Perafán, 2017*), we speculate that their venom DRPs exhibit relatively lower sequence diversity (*Figure 5*, *Figure 5—source data 1*). In complete contrast, venom DRPs in mygalomorph spiders that mostly rely on venom and not silk, being either ambush or sit-and-wait predators to capture a much diverse prey base, appear to have experienced a significantly greater influence of the diversifying selection (*Beydizada et al., 2022*; *Figure 5*, *Figure 5—source data 1*). However, the current literature and our investigation are limited to the most diverse lineage in Araneomorphae - the RTA clade. Since this lineage does not employ silk webs for predation, Scenario 3 is unlikely to explain the current observations. Surprisingly, however, despite being the most speciose spider lineage, and having a significantly higher genomic diversification rate in comparison to other araneomorphs (*Fernández et al., 2018*), the lack of toxin sequence diversity in the RTA clade is intriguing (*Figure 5*, *Figure 5—source data 1*). It should also be noted that venom toxins from the foraging web-building araneomorphs outside the RTA clade are very poorly studied (e.g. only a handful of species are investigated from a biodiscovery perspective, and not a single toxin has been sequenced at the nucleotide level to date).

## Deployment dictates spider venom evolution

The current literature is replete with findings that support the strong influence of positive selection on genes encoding venom toxins in diverse animal lineages (*Juárez et al., 2008*; *Sunagar et al., 2012*; *Sunagar et al., 2013*; *Župunski and Kordiš, 2016*). Venom proteins are theorised to follow a 'two-speed' mode of evolution, wherein they readily diversify in animals that experience drastic shifts in ecology and/or environment - a prominent feature of evolutionarily younger lineages [e.g. cone snails and advanced snakes with evolutionary origins dating back to <35–50 MYA (*Sunagar and Moran, 2015*)]. This rapid expansion, or the 'expansion phase', is shaped by a strong influence of positive selection that underpins the transition of organisms into novel ecological niches. Post these adaptive changes, the influence of diversifying selection is replaced by the effects of purifying selection (the 'purification phase') that preserve potent toxins generated during the expansion phase. This, perhaps, explains the contrasting evolutionary regimes documented in evolutionarily younger and ancient lineages (*Sunagar and Moran, 2015*). Venom coding genes in evolutionarily ancient lineages are said

to re-enter the expansion phase if they re-encounter dramatic shifts in ecology and environment. The only exceptions to this hypothesis are toxins that non-specifically interact with their molecular targets or those that are deployed for antipredatory defence (*Sunagar and Moran, 2015*). The latter hypothesis, however, mostly stems from the analyses of venom proteins that are deployed for predation. A dearth of sequence information for venom components majorly employed for antipredator defence has impeded our understanding of their evolutionary diversification.

Spiders of the genera, *Hadronyche* and *Atrax* (family Atracidae), are known to deploy their DRP toxin superfamily (SF13: Ares) predominantly for antipredatory defence (*Herzig et al., 2020*). In contrast, tarantulas of the family Theraphosidae are known to mostly employ their venom to capture prey animals. This provided us with a unique opportunity to comparatively investigate the molecular evolution of spider venom proteins chiefly deployed for predation (SF74, SF77, SF79, SF89, SF90, SF92, and SF99 from Theraphosidae) and self-defence (SF13 from Atracidae). Our analyses of the molecular evolutionary histories of theraphosid spider venom DRPs deployed for prey capture reveal a strong influence of diversifying selection ($\omega$: 1.2–2.9; PS: 0–11; *Figure 5—source data 1*, *Figure 5—figure supplement 1*), whereas those employed for self-defence in Atracidae spiders were constrained by negative selection ($\omega$: 0.8; PS: 3; *Figure 5—source data 1*, *Figure 5—figure supplement 1*). Outcomes of FUBAR and MEME analyses further corroborated these findings. FUBAR identified numerous sites (~10%) in defensive toxins as evolving under the pervasive influence of negative selection, while MEME detected several episodically diversifying sites (~22%) in theraphosid toxins deployed for prey capture (*Figure 5—source data 1*).

Such contrasting modes of diversification could be attributed to the 'two-speed' mode of venom evolution, where the offensive toxins gain an evolutionary advantage over prey by amplifying their sequence and functional diversity (*Sunagar and Moran, 2015*). In contrast, as defensive venoms are infrequently deployed, or have evolutionarily conserved molecular targets across predatory lineages, they experience relatively reduced effects of diversifying selection. In the absence of a need for sequence variation, purifying selection pressures instead ensure the preservation of broadly effective toxins (*Sunagar and Moran, 2015*).

## Methods

### Sequence data curation and assembly

Nucleotide datasets consisting of Mygalomorphae DRP sequences were assembled from the National Center for Biotechnology Information's Non-redundant and Transcriptome Shotgun Assembly databases using manual search and exhaustive BLAST iterations (*Altschul et al., 1990*). Sequences for Araneomorphae toxins were retrieved using a similar strategy, while additional sequences from the RTA clade were derived from *Cole and Brewer, 2021*. A list of sequence data analysed in this study has been provided as *Figure 4—source data 1*. Translated sequences were aligned in MEGA X (v 10.2.6) using MUSCLE (*Edgar and Batzoglou, 2006*; *Kumar et al., 2018*) before back-translation to nucleotides. Alignment was further refined by using structurally conserved cysteines as guides.

### Toxin superfamily identification and nomenclature

Spider toxin superfamilies were identified based on the strong conservation observed in signal peptide and propeptide sequences as illustrated by *Pineda et al., 2020*; *Pineda et al., 2014*. Additional support for uniqueness of each superfamily was obtained by our extensive phylogenetic (both Bayesian and Maximum likelihood) and Principal Component Analyses of toxin sequences. All the novel toxin superfamilies were labelled after gods/deities of death, destruction and underworld based on a nomenclature system as described before (*Pineda et al., 2014*). A list of novel superfamilies identified has been included as a part of supplementary file (*Supplementary file 1*).

### Phylogenetic analyses

Phylogenetic histories of toxin families were reconstructed for whole length toxin nucleotide sequence using Bayesian and maximum-likelihood inferences implemented in MrBayes 3.2.7 a (*Altekar et al., 2004*; *Ronquist et al., 2012*) and IQ-TREE v1.6.12 (*Nguyen et al., 2015*; *Chernomor et al., 2016* f), respectively. All alignments utilised have been made available in *Figure 2—source data 1*, *Figure 3— source data 1* and *Figure 4—source data 1*, respectively. Bayesian analyses were run for a minimum

of ten million generations using twelve Markov chains across four runs, sampling every 100th tree. Twenty-five percent of the total trees sampled were discarded as burn-in. The log-likelihood score for each tree was plotted against the number of generations to assess whether the analysis has reached an asymptote. A stop value of 0.01 was used for the average standard deviation of split frequencies. Bayesian Posterior Probability (BPP) was used to evaluate node support for the branches of Bayesian trees. ML analyses were performed using IQ-TREE with an edge-proportional partition model and 100 Bootstrap replicates. The best partition scheme for each partition was determined by utilising the inbuilt ModelFinder plugin in IQ-TREE (*Kalyaanamoorthy et al., 2017*). Phylogenetic trees were rooted with non-venom nucleolar cysteine-rich protein sequences from *Mastigoproctus giganteus*, *Stenochrus portoricensis*, *Prokoenenia wheeleri*, *Phrynus marginemaculatus* and *Cryptocellus centralis* from the class Arachnida that fall outside of the suborder Opisthothelae.

## Principal component analysis

PCA of signal peptide sequences from spider toxin superfamilies was performed in R (v 4.1.2; *R Development Core Team, 2021*) using a previously published script (*Konishi et al., 2019*; https://github.com/TomokazuKonishi/direct-PCA-for-sequences; (*McClellan and McCracken, 2001*). Sequences were aligned using MUSCLE in MEGA X (v 10.2.6) (*Edgar and Batzoglou, 2006*; *Kumar et al., 2018*) and further digitising in R utilising boolean vectors. The scaled principal component values (sPC) were calculated using conventional PCA prior to plotting.

## Assessment of molecular evolution

The nature of selection shaping the evolution of DRP toxins was determined using a maximum-likelihood inference implemented in CodeML of the PAML package (v 4.9 j) (*Yang, 2007*). Super-families with a minimum of 15 sequence representatives were further down-selected for this analysis to avoid inaccurate estimation of omega values when analysing smaller datasets. The ratio of non-synonymous substitutions (nucleotide changes that alter the coded amino acid) to synonymous substitutions (nucleotide changes that do not alter the coded amino acid), also known as omega ($\omega$), was estimated. A likelihood ratio test (LRT) for the nested models - M7 (null model) and M8 (alternate model) - was performed to assess the statistical significance of the findings. The Bayes Empirical Bayes (BEB) approach implemented in M8 was used to calculate the posterior probabilities for site classes (*Yang et al., 2005*). Amino acid sites with a posterior probability of over 95% (PP ≥95%) were inferred as positively selected. The episodic and pervasive nature of selection was determined using the Mixed Effect Model of Evolution (MEME; *Murrell et al., 2012*) and the Fast Unconstrained Bayesian AppRoximation (FUBAR; *Murrell et al., 2013*), respectively.

## Evaluation of selection on amino acid properties

The influence of positive selection on the biochemical and structural properties of amino acids was evaluated using TreeSAAP (v 3.2; *Woolley et al., 2003*). TreeSAAP estimates the rate of selection using a modified MM01 model (*McClellan and McCracken, 2001*). Statistical probabilities corresponding to a range of properties were further calculated for each amino acid. BASEML was set to run with the REV model and eight evolutionary pathway categories were defined for evolutionary pathway analyses with a sliding window size set to one. Data acquired from TreeSAAP was further visualised and processed with IMPACT_S (*Maldonado et al., 2014*).

## Structural analyses

Structural homologues of spider toxin superfamilies were identified via blast searches against the RCSB Protein Data Bank (https://www.rcsb.org/) and subsequently modelled using the SWISS-MODEL web server via user template mode (*Waterhouse et al., 2018*). The resultant models were validated using MolProbity (v 4.4; https://github.com/rlabduke/MolProbity; *Williams et al., 2018*) and general Ramachandran plot. Regimes of evolutionary selection pressures were evaluated and mapped onto homology models using the Consurf webserver (*Ashkenazy et al., 2016*, http://consurf.tau.ac.il/).

PyMOL v2.5.2 (Schrödinger, LLC, USA) was used to visualise and generate the images of homology models.

## Acknowledgements

This work was supported by the DBT/Wellcome Trust India Alliance Fellowship [grant number IA/I/19/2/504647] awarded to KS. NYS is thankful to Vivek Suranse and Senji Laxme R R (Indian Institute of Science) for insightful discussions.

## Additional information

### Funding

| Funder | Grant reference number | Author |
| --- | --- | --- |
| The Wellcome Trust DBT India Alliance | IA/I/19/2/504647 | Kartik Sunagar |

The funders had no role in study design, data collection and interpretation, or the decision to submit the work for publication. For the purpose of Open Access, the authors have applied a CC BY public copyright license to any Author Accepted Manuscript version arising from this submission.

### Author contributions

Naeem Yusuf Shaikh, Conceptualization, Data curation, Formal analysis, Investigation, Visualization, Methodology, Writing - original draft; Kartik Sunagar, Conceptualization, Resources, Formal analysis, Supervision, Funding acquisition, Validation, Investigation, Visualization, Methodology, Writing - original draft, Project administration, Writing - review and editing

### Author ORCIDs

Naeem Yusuf Shaikh ⓘ http://orcid.org/0000-0002-9903-8484
Kartik Sunagar ⓘ http://orcid.org/0000-0003-0998-1581

### Decision letter and Author response

Decision letter https://doi.org/10.7554/eLife.83761.sa1
Author response https://doi.org/10.7554/eLife.83761.sa2

## Additional files

### Supplementary files

• Supplementary file 1. List of spider toxin superfamilies. The table lists spider toxin superfamilies described till date with names conferred upon and their respective mythological references. A list of genera from which the superfamily was identified has also been provided.

• MDAR checklist

### Data availability

All data generated or analysed during this study are included in the manuscript and supporting files. Source data files have been provided for Figures 2, 3, 4 and 5.

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
