## [Editor Report]

This is an important survey of disulfide-rich peptides (DRPs), which comprise a large fraction of the most functionally important components of spider venom. While spider DRPs were thought to have evolved independently numerous times throughout the spider tree of life, the authors make a solid case for the idea that they all stem from a single common ancestral protein. The study makes a significant advance towards formalizing the diversity of spider venoms, which will be of interest both to scientists working on protein evolution and to those working on functional venomics.

---

## [Decision Letter]

**Decision letter after peer review:**

Thank you for submitting your article "The primordial knot: the deep-rooted origin of the disulfide-rich spider venom toxins" for consideration by *eLife*. Your article has been reviewed by 3 peer reviewers, including Ariel Chipman as Reviewing Editor and Reviewer #1, and the evaluation has been overseen by Christian Landry as the Senior Editor. The following individual involved in the review of your submission has agreed to reveal their identity: Michael Brewer (Reviewer #3).

Essential revisions:

All three reviewers judge that this is an important body of work. However, in the discussion there was also agreement that some claims need to be better substantiated and that some of the analyses will need to be re-done unless you can make a convincing case for why they were done as they were. Please note the following key points:

1) Explain why not all the data from the cited references were included (most notably ref. 38). If you cannot justify not including all the data, please repeat the analysis with all available data.

2) Modify your conclusions to be specific about the representation of spider diversity included in the study.

3) Give more details in the Methods section to make your conclusions stronger and better supported.

4) Correct errors in phylogenetic nomenclature and usage of terminology (e.g. confusion between clade names and ancestors).

5) Explain the names and references from mythology and popular culture to make them clear to a culturally diverse readership.

*Reviewer #1 (Recommendations for the authors):*

The manuscript is rich in cultural references and metaphors. This is not a problem on its own, but in many cases, the reason for the metaphor or cultural reference is unclear. I am fully in favor of authors bringing in their own cultural background, but it should be put in context for the reader. I assume the "ancestral knot" in the manuscript's title refers to knottin proteins, but this is not made clear, and I may be missing a deeper reference. The decision to name the ancestral protein "Adi Shakti" is confusing until one realizes that all the protein super-families were given names from Indian mythology (and also a few from other mythologies?) Are these names that were given by the authors as part of the preparation of this paper, or did some of them already exist? Is there a convention to name venom proteins after figures from Indian mythology? I don't know, and I assume the reader won't either. Please give this background explicitly somewhere. Similarly, is there a deeper context to "many ways to skin a cat"? There might be something I am missing, but this seems to me to be a metaphor out of place. This is a short paper, you can afford to expand a bit on some of the metaphors and references.

On a completely different note – while the results are interesting and important, there is a bit too much use of hyperbolic language. I suggest cutting a few "fascinating" and "astonishing".

*Reviewer #2 (Recommendations for the authors):*

This interesting study looks into the evolution of putative spider venom toxins, specifically disulfide-rich peptides. The results and interpretations presented are a promising start, but I don't consider them to be sufficiently convincing yet to warrant publication in *eLife* in their current state. Below I make suggestions for some further analyses that can make the conclusions more robust.

The study builds upon previous work, especially references 9, 12, and 38. However, I think that the authors need to discuss more clearly how their results and conclusions relate to those reached in these papers. They should do this on several levels to enable readers, who are not spider venom experts, to better understand this contribution. For instance, on the most superficial level, it would be helpful to better explain that the strategy of naming toxin superfamilies for deities, which struck this reviewer as peculiar, was actually established in previous works. More importantly, clarification is needed about the sources of the sequences. The paper discusses the results from the perspective of spider venom. Yet, all the non-ctenid araneomorph data seem to be derived from whole-body transcriptomes generated by reference 38. Therefore, the results have to be presented with caution because the tissue source of the putative toxin sequences in these taxa is uncertain.

Moreover, it is difficult for me to understand how the authors identified new DRP superfamilies when the phylogenetic analyses that they provide do not seem to incorporate all previously published relevant data. For instance, of the 26 DRP superfamilies identified in reference 9, only 14 are included in the analyses presented here. This makes it impossible to assess how these relate to the new superfamilies reported here, or if some of the new superfamilies may be a subset of previously established superfamilies or vice versa. Why were these data excluded from the phylogenetic analyses? This is puzzling because they are included in the supplementary alignment S10. It makes sense to me to also include these excluded superfamilies in the phylogenetic analyses.

– I suggest removing the reference to Adi Shakti from the abstract or including an explanation because few non-Hindus will understand what this refers to.

– This study attempts to trace the evolutionary origin of DRPs in a phylogenetically diverse clade with many spider genera and families. It would be a very helpful context for readers if the authors clarify on a schematic phylogeny of spiders exactly from which genera and families the data in their paper comes and which higher-level clades these represent. For instance, figure 2 includes genera from several araneomorph families. Which families? This gives readers a better sense of how comprehensively the analyses data represent spider phylogeny as a whole.

– Figures 1 and 2; please insert clade support values. Also please clarify if whole toxin sequences or just signal and propeptide sequences were used to construct the trees. Also, which of the included superfamilies are new? Indicate that in the tree or legend and amend the title. Also, where did the data for the taxon Pisaurina come from? It is not present in reference 38, which is mentioned as the source of the araneomorph sequences.

– Figure 3B: this alignment suggests that there are no gaps. Is that correct? If not, please indicate alignment gaps.

– Lines 90-93: it is more accurate to also mention that the cysteine framework was used to identify superfamilies as well.

– Lines 140-141: Opisthothelae denotes a clade comprising araneomorphs and mygalomorphs, not their common ancestor, so the analysis traces DRPs to the origin of this clade.

– Line 145: 'speculate on'.

– Lines 188-189: you refer to figure 3a when mentioning Adi Shakti. I don't understand how this relates to the figure. Do you want to call the ancestral cysteine scaffold Adi Shakti? If so, say this specifically. And also, what does this ancestral scaffold look like? References 9 and 38 reconstruct the ancestral DRP as having 8 cysteines. Do your analyses agree with this? Was this tested?

– Lines 189-192: this should be rephrased because there certainly are other spider venom toxins that do not belong to the DRP group.

– Lines 204-205: rephrase. You have reconstructed the origin of DRPs in the last common ancestor of Opisthothelae, which is a clade, not an ancestor.

– Line 208: positively selected sites or amino acids.

– Line 224: predominate in the venoms.

– Lines 238-240 (also lines 128-130, 262-264): I am unconvinced by the arguments here and by the data claimed to support this. Genera and families are Linnean ranks of arbitrary phylogenetic depth. They do not meaningfully coincide with biologically relevant lineages. Moreover, the taxa sampled only cover a small fraction of the phylogenetic breadth of spider lineages. As a result, interpretational difficulties emerge. For example, in Figure 1 some of the genera represent families (Trittame is the only taxon representing Barychelidae), and some genera are not split (Atrax and Hadronyche are colour coded the same, and together represent the family Atracidae), while in Figure 2 several of the supposed family-level taxa are only represented by a single genus, including Salticidae, Thomisidae, Lycosidae, and Homalonychidae. It, therefore, becomes impossible in many instances to distinguish whether toxin recruitments happened on the level of genera, families, or even higher-level taxa. To use Figure 1 to argue that mygalomorph toxin recruitments happened "post divergence of family members" (lines 263) is unconvincing since the included genera represent just three families that diverged between 150-100 million years ago (see Opatova et al., 2019, Syst. Biol. 69: 671). To get a more accurate estimate of when toxin families were recruited the authors would have to do ancestral state reconstruction on an established phylogeny.

– Lines 265-266: I don't understand this. Do the araneomorphs have fewer DRP superfamilies than the mygalomorphs?

– Line 329: were the phylogenetic analyses done on the basis of nucleotide or amino acid data? Also, what substitution models were used to generate the trees, and how were these chosen?

– Line 393: correct spelling of Mygalomorphae.

– Line 415: the dotted line is black.

– Figure 5: you need to reverse the colour labeling for 3FTXs and SVMPs, which should be blue and green, respectively, and you need to pair the 3FTXs with the elapids and the SVMPs with the vipers. Also, do the donut charts depict proteomic or transcriptomic data, and are they for specific species (if so, which?) or some sort of average (if so, based on what species and how were the proportions calculated?)? Moreover, in contrast to what you write in the legend and main text (e.g. lines 227-228), it seems that for each of the major groups of venomous snakes one or two types of toxins dominate venom composition as well, just like in the spiders. Please discuss this.

– lines 228-229: this is also true for spiders. Their venoms do not exclusively contain DRPs.

– line 243 and further (and related text in the Results section): the contrast between mygalomorphs and araneomorphs in terms of the evolutionary selection pressures governing toxin evolution is indeed striking at first sight. However, before trying to explain these in general terms (lines 261 and further), the authors need to ask if these results can actually be generalized. 11 of the 12 identified superfamilies of mygalomorph toxins that evolve under the influence of positive selection are limited to just two genera: Hadronyche and Haplopelma. How representative are just these two genera for conclusions about mygalomorph evolution generally? Are the results not equally well explained (or better) in terms of the specific biology of these two taxa? Similarly, 16 of the 20 DRP superfamilies in the selection analyses for araneomorphs are restricted to just 1 araneomorph family (Ctenidae), which detracts from how general the results can be claimed to be. Moreover, it is not clear to me why the selection analyses were performed on only a subset of the spider DRPs included in this study and previously published work. How were these superfamilies selected, and why were others excluded? Could the picture change if more superfamilies were analysed? This makes it impossible to judge how general the results may be as indicators for evolutionary mechanisms in the two clades.

– Figure S2: I don't understand the significance of the 'novel' cysteines in green. Are these depicted separately because they are part of the scaffold? Please clarify. Also, are the names given to these new superfamilies, like Seker, the official new superfamily names you propose or are they there to enliven the trees? Please clarify. Also, make clear that these exclude the novel non-DRP toxins.

– Supplementary figures 5, 6, and 9: please make sure that the published trees have legible clade support measures.

*Reviewer #3 (Recommendations for the authors):*

I greatly enjoyed reading your work and am grateful to be asked to review it. I hope my comments improve your already well-done study.

Please see the comments in my public review regarding method documentation. Perhaps a GitHub repository documenting all of the analyses and code along with a Figshare repository containing all of the data, results, and supplements would be enough.

I hope I did not misunderstand your use of only our ctenid data to represent araneomorphs. I hope the example datasets I provided are useful, and I apologize if my interpretation is incorrect.

Targeted comments:

The abstract states that "these lineages constitute over 50% of the extant spiders" following a discussion of the Mygalomorphae and Araneomorphae. These two lineages comprise far more than 50% of spider species. Are you referring to the Mygalomorphae + RTA-clade only?

Line 63 – "primitive Mesothelae species" Species are not primitive, but individual traits can be.

Lines 118 and 124: "monophyletic clade" is redundant.

Lines 236 – 241: Is the comparison of taxonomic ranks employed here fair? Are spider families and genera separated by 350 million years assumed to be comparable? Perhaps evolutionary ages or relative species diversity within the ranks could standardize the comparisons. I am not sure how best to approach this, but it stands out to me as potentially problematic, but I am not sure it greatly impacts your overall conclusions.

Lines 264 – 266: While it is true that many araneomorphs rely on prey capture webs, none of the species included in this work do. Making these claims without including species that do utilize prey-capture webs is tenuous.

I greatly enjoyed this work and think it will advance the field forward. I am honored that you utilized the data presented in my former PhD student's and my work so prominently and appreciate the novel directions you took with them. I hope my comments are useful and look forward to seeing a future version of the manuscript.

---

## [Author Response]

Essential revisions:All three reviewers judge that this is an important body of work. However, in the discussion there was also agreement that some claims need to be better substantiated and that some of the analyses will need to be re-done unless you can make a convincing case for why they were done as they were. Please note the following key points:1) Explain why not all the data from the cited references were included (most notably ref. 38). If you cannot justify not including all the data, please repeat the analysis with all available data.

We have, indeed, analysed all spider toxin sequences available to date. We have relied on the signal and propeptide regions for identifying novel superfamilies, which is an accepted convention: Pineda et al. (2014 BMC Genomics); Pineda et al. (2020 PNAS). Although many additional superfamilies can be identified, we have only retained those sequences for which there were at least 5 representatives for the identification of toxin superfamilies, and 15 representatives for selection analyses to ensure robustness. This filtering step ensured that the generated alignments, phylogenetic trees and evolutionary assessments were robust and devoid of noise that stems from single-representative groups. Adding in those sequences would have enabled us to identify many more superfamilies, solely based on the signal and propeptide examination, but it wouldn’t have been possible to support them with other lines of evidence that were provided for all other superfamilies in this study, jeopardising the overall quality of the manuscript. Nonetheless, there is strong evidence that the left-out sequences are also related to the ones analysed in this study (Figure 4 —figure supplement 2). In future, when more transcriptomes are sequenced, it would be possible to designate these newer toxin superfamilies with much stronger support.

Please note that, in response to some of the reviewer’s comments, we have now included additional sequences and reanalysed data, which has now led to the identification of 5 additional toxin superfamilies.

2) Modify your conclusions to be specific about the representation of spider diversity included in the study.

We were attempting to explain the contrasting rates of diversification of toxin superfamilies in mygalomorphs and araneomorphs analysed in this study, as providing no explanation for this very interesting trend would have left the readers wondering. We had listed all possible scenarios that explain the observed trend, including the ability to spin webs by the araneomorphs. However, we did clarify that this explanation lacks support at this stage since we have only analysed sequences from non-web-building RTA clade.

Line 371: “However, the current literature and our investigation are limited to the most diverse lineage in Araneomorphae – the RTA clade.”

We understand now that this can be confusing to the reader. Hence, we have completely modified the title of this section to “*Distinct recruitment and diversification of spider venom superfamilies in Mygalomorphae and Araneomorphae*”. Moreover, we have also analysed an additional toxin superfamily from web-building spiders (the only one in the literature for which there are sequences available), and the outcome of this analysis supports our hypothesis. However, since toxin sequences belonging to diverse DRP superfamilies from web-building araneomorphs (amino acids or nucleotides) have not been sequenced or deposited to date, this hypothesis has limited support at this stage. We have made this very clear in the revised version of the manuscript. We have also provided alternative explanations for observing this contrasting trend, and avoided highlighting the web-building vs non-web-building spider comparisons throughout the paper (now only discussed as one of the three hypotheses in the discussion).

3) Give more details in the Methods section to make your conclusions stronger and better supported.

We have now further expanded our methods section for better reproducibility and have included all the possible details.

4) Correct errors in phylogenetic nomenclature and usage of terminology (e.g. confusion between clade names and ancestors).

We thank the reviewers for their valuable suggestions regarding nomenclature. We apologise for the inconsistency in the usage of some of the terminologies. We have now made appropriate changes.

5) Explain the names and references from mythology and popular culture to make them clear to a culturally diverse readership.

We have now included a supplementary table describing the names and their cultural references (Supplementary File 1).

Reviewer #1 (Recommendations for the authors):The manuscript is rich in cultural references and metaphors. This is not a problem on its own, but in many cases, the reason for the metaphor or cultural reference is unclear. I am fully in favor of authors bringing in their own cultural background, but it should be put in context for the reader. I assume the "ancestral knot" in the manuscript's title refers to knottin proteins, but this is not made clear, and I may be missing a deeper reference. The decision to name the ancestral protein "Adi Shakti" is confusing until one realizes that all the protein super-families were given names from Indian mythology (and also a few from other mythologies?) Are these names that were given by the authors as part of the preparation of this paper, or did some of them already exist? Is there a convention to name venom proteins after figures from Indian mythology? I don't know, and I assume the reader won't either. Please give this background explicitly somewhere. Similarly, is there a deeper context to "many ways to skin a cat"? There might be something I am missing, but this seems to me to be a metaphor out of place. This is a short paper, you can afford to expand a bit on some of the metaphors and references.On a completely different note – while the results are interesting and important, there is a bit too much use of hyperbolic language. I suggest cutting a few "fascinating" and "astonishing".

Spider toxin superfamilies have been named after gods/deities of death, destruction and the underworld based on nomenclature introduced by Pineda et al. (2014 BMC genomics). This convention was also followed in subsequent papers introducing novel toxin superfamilies Pineda et al. (2020 PNAS). We have attempted to include names from diverse cultures, and not only named our new superfamilies after Hindu gods and goddesses, but also from Egyptian, Greek, and other mythologies. We have now included this explanation in the manuscript under the methods and Results sections. We have also provided additional details pertaining to this nomenclature in Supplementary File 1. We have also left out unclear references (e.g., many ways to skin a cat) and reinforced efforts to expand key references (such as: Adi Shakti, knottin, etc.) to make this manuscript more reader friendly.

Reviewer #2 (Recommendations for the authors):This interesting study looks into the evolution of putative spider venom toxins, specifically disulfide-rich peptides. The results and interpretations presented are a promising start, but I don't consider them to be sufficiently convincing yet to warrant publication in eLife in their current state. Below I make suggestions for some further analyses that can make the conclusions more robust.

We thank the reviewer for their constructive inputs and suggestions. We hope that our responses have met the reviewers’ expectations.

The study builds upon previous work, especially references 9, 12, and 38. However, I think that the authors need to discuss more clearly how their results and conclusions relate to those reached in these papers. They should do this on several levels to enable readers, who are not spider venom experts, to better understand this contribution. For instance, on the most superficial level, it would be helpful to better explain that the strategy of naming toxin superfamilies for deities, which struck this reviewer as peculiar, was actually established in previous works.

We thank the reviewer for their suggestion. Novel Spider toxin superfamilies identified in this study were named after gods/deities of death, destruction and the underworld based on nomenclature as introduced by Pineda S, et al. (2014, BMC genomics; 2020, PNAS). We have now included a list of names with appropriate cultural references in the supplementary text (Supplementary File 1) and also clarified this nomenclature in the methods.

We have not tested the four-disulfide bond stabilised ancestral DRP hypothesis as theorised by Pineda et al. (2020 PNAS) and Cole and Brewer (2020 bioRxiv). It would be difficult to trace the exact nature and type of this ancestral scaffold given the long evolutionary history and their molecular divergence. Multiple events of duplication and diversification throughout their evolutionary history have resulted in events of gain and loss of cysteines and functional residues. As a result, we find diverse DRP scaffolds exhibiting a range of cysteine pairs – 2 to 6 disulfide bonds (i.e 4 to 12 cysteines) within the same spiders at times (e.g., an individual spider from the *Haplopelma* genus may contain SF90, SF91, SF99 and SF100 toxin superfamilies in its venom with 6, 8, 12 and 10 cysteines, respectively). In our opinion, this could be only identified by the examination of the genomic architectures (synteny) which is beyond the scope of this study. Our extensive phylogenetic and evolutionary analyses provide a deeper insight into the common origin of these DRP toxins in spiders dating as old as the split of Mygalomorphae and Araneomorphae, but we cannot comment on the nature and type of those scaffolds.

More importantly, clarification is needed about the sources of the sequences. The paper discusses the results from the perspective of spider venom. Yet, all the non-ctenid araneomorph data seem to be derived from whole-body transcriptomes generated by reference 38. Therefore, the results have to be presented with caution because the tissue source of the putative toxin sequences in these taxa is uncertain.

We would like to clarify that the non-ctenid araneomorph sequences utilised in this study were identified as putative toxins based on homology and DRP motif conservation (Cole and Brewer 2021, bioRxiv). This is further supported by our sequence alignments and phylogenetic analyses, wherein these toxin sequences are nested with annotated and experimentally characterised toxin sequences (Figures 3, 4 and respective figure supplements). We have also modified our methods section with regards to clarification of sequence data used. We previously included a supplementary dataset (now Figure 4 – Source data 2) with all sequences analysed in this study.

Moreover, it is difficult for me to understand how the authors identified new DRP superfamilies when the phylogenetic analyses that they provide do not seem to incorporate all previously published relevant data. For instance, of the 26 DRP superfamilies identified in reference 9, only 14 are included in the analyses presented here. This makes it impossible to assess how these relate to the new superfamilies reported here, or if some of the new superfamilies may be a subset of previously established superfamilies or vice versa. Why were these data excluded from the phylogenetic analyses? This is puzzling because they are included in the supplementary alignment S10. It makes sense to me to also include these excluded superfamilies in the phylogenetic analyses.

We had previously described the criteria that was used to define SFs in the Results section (line 85). We have relied on the signal and propeptide regions for identifying novel superfamilies, which is an accepted convention: Pineda et al. (2014 BMC Genomics); Pineda et al. (2020 PNAS). Evidence was further generated to support the monophyly of these toxin superfamilies using various means, including phylogenetics, sequence clustering and PCA. The excluded superfamilies do not have the same signal and propeptide regions as any other superfamily described here. Hence, it is just not possible that they are a subset of the superfamilies described here. In support of this, we previously included a supplemental alignment Figure S10 (now Figure 4 —figure supplement 2) of all toxin superfamily sequences analysed in this study.

Line 85 “Superfamilies (SF) of venom toxins in spiders have been classified based on their signal peptide and propeptide sequences (12)”

The reason for excluding certain previously described superfamilies from our phylogenetic analysis was the lack of nucleotide or amino acid sequence data. Adding in such highly divergent sequences, where there aren’t enough representatives, only adds phylogenetic noise and prevents the generation of robust phylogenies. As explained above, because of the differences in signal and propeptide regions, it is just not possible that the excluded superfamilies are a part of any of the superfamilies being described here. Therefore, adding them to our phylogenies would have only presented their relative position in the tree. We have, indeed, attempted to build such large phylogenies where every sequence was included. However, as explained above, because of the extreme sequence divergence in these toxins and the lack of enough representative sequences from some of the superfamilies, the topology of the tree was completely unresolved.

– I suggest removing the reference to Adi Shakti from the abstract or including an explanation because few non-Hindus will understand what this refers to.

While we completely agree that this may not be very clear to many, including many Hindus as it is a Sanskrit world, we feel that it is an important reference to the primordial origin of spider venoms. Following the reviewer’s suggestion, we have provided clarity on this in the abstract.

– This study attempts to trace the evolutionary origin of DRPs in a phylogenetically diverse clade with many spider genera and families. It would be a very helpful context for readers if the authors clarify on a schematic phylogeny of spiders exactly from which genera and families the data in their paper comes and which higher-level clades these represent. For instance, figure 2 includes genera from several araneomorph families. Which families? This gives readers a better sense of how comprehensively the analyses data represent spider phylogeny as a whole.

We thank the reviewer for this valuable suggestion. We have now introduced Figure 1 which depicts the schematic phylogeny of spiders and the sampled genera.

– Figures 1 and 2; please insert clade support values. Also please clarify if whole toxin sequences or just signal and propeptide sequences were used to construct the trees. Also, which of the included superfamilies are new? Indicate that in the tree or legend and amend the title. Also, where did the data for the taxon Pisaurina come from? It is not present in reference 38, which is mentioned as the source of the araneomorph sequences.

We have now made appropriate changes to the figure making it more informative. We have used the whole toxin sequence to construct phylogenies and this has been clearly described in the methods now. We did not add node support values to the trees as they are unreadable and uninformative. Instead, we have used thick dark (BPP>0.95; ML>90) and light grey (BPP<0.95; ML<90) branches to indicate node support. This is now clearly indicated in the legend of all phylogenetic figures.

We have retrieved Pisaurina DRP toxin sequences from the dataset by Cole and Brewer (2021, bioRxiv, Ref. 38). We request the reviewer to refer to Table 4 of that paper.

– Figure 3B: this alignment suggests that there are no gaps. Is that correct? If not, please indicate alignment gaps.

In this schematic representation (not an alignment), we intend to highlight the strongly conserved cysteine framework. The actual alignment can be viewed in Figure 4 —figure supplement 2. We have now further elaborated on this in the figure legend.

– Lines 90-93: it is more accurate to also mention that the cysteine framework was used to identify superfamilies as well.

We did not employ the cysteine framework to identify superfamilies. Superfamilies were identified using signal and propeptide as mentioned above. The criteria used to define SFs was previously described under the Results section (line 85). We have now further expanded on this to provide more clarification. The cysteine framework was rather used to highlight the homology across spider venom toxin superfamilies.

Line 85 “Superfamilies (SF) of venom toxins in spiders have been classified based on their signal peptide and propeptide sequences (12)”

– Lines 140-141: Opisthothelae denotes a clade comprising araneomorphs and mygalomorphs, not their common ancestor, so the analysis traces DRPs to the origin of this clade.

We thank the reviewer for pointing out this oversight. We have now corrected it to suborder Opisthothelae as the clade encompassing Mygalomorphae and Araneomorphae spiders.

– Line 145: 'speculate on'.

We have incorporated the suggested change.

– Lines 188-189: you refer to figure 3a when mentioning Adi Shakti. I don't understand how this relates to the figure. Do you want to call the ancestral cysteine scaffold Adi Shakti? If so, say this specifically. And also, what does this ancestral scaffold look like? References 9 and 38 reconstruct the ancestral DRP as having 8 cysteines. Do your analyses agree with this? Was this tested?

We apologise for the confusion caused. We confer the name ‘Adi Shakti’ to the ancestral DRP scaffold suggesting a deep-rooted origin. We have made appropriate changes to the text under the Discussion section to reflect this with more clarity:

Line 227 “Our findings strongly suggest a deep-rooted origin of DRP spider venom toxin superfamilies (Figure 4A), possibly from a single ancestral DRP or knottin scaffold, which we name ‘Adi Shakti’, after the original creator of the universe according to Hindu mythology”.

We have not tested the four disulfide bond stabilised ancestral DRP hypothesis as theorised by Pineda et al. (2020 PNAS). It would be difficult to trace the exact nature and type of this ancestral scaffold given the long evolutionary history and their molecular divergence. Multiple events of duplication and diversification throughout their evolutionary history have resulted in events of gain and loss of cysteines and functional residues. As a result, we find diverse DRP scaffolds exhibiting a range of cysteine pairs – 2 to 6 disulfide bonds (i.e., 4 to 12 cysteines) within the same spiders at times (e.g., an individual spider from the *Haplopelma* genus may contain SF90, SF91, SF99 and SF100 toxin superfamilies in its venom with 6, 8, 12 and 10 cysteines, respectively). In our opinion, this could be only identified by the examination of the genomic architectures (synteny) which is beyond the scope of this study. Our extensive phylogenetic and evolutionary analyses provide a deeper insight into the common origin of these DRP toxins in spiders dating as old as the split between Mygalomorphae and Araneomorphae, but we cannot comment on the nature and type of those scaffolds.

– Lines 189-192: this should be rephrased because there certainly are other spider venom toxins that do not belong to the DRP group.

We have now modified the statement appropriately to distinctly focus on DRP toxin superfamilies.

– Lines 204-205: rephrase. You have reconstructed the origin of DRPs in the last common ancestor of Opisthothelae, which is a clade, not an ancestor.

We thank the reviewer for pointing out this oversight. We have now corrected it to suborder Opisthothelae as the clade encompassing Mygalomorphae and Araneomorphae spiders.

Line 244: “… originated from an ancestral scaffold in Opisthothelae, the clade that encompasses Mygalomorphae and Araneomorphae spiders”.

– Line 208: positively selected sites or amino acids.

We thank the reviewer for their suggestion and have incorporated the suggested change.

Line 249: “Consistent with this hypothesis, the majority of positively selected sites (~96%) identified in spider venom DRP toxins …”

– Line 224: predominate in the venoms.

We have incorporated the suggested change and modified the statement accordingly.

– Lines 238-240 (also lines 128-130, 262-264): I am unconvinced by the arguments here and by the data claimed to support this. Genera and families are Linnean ranks of arbitrary phylogenetic depth. They do not meaningfully coincide with biologically relevant lineages.

We respectfully disagree with this argument. Certain adaptive traits (e.g., the origin of potent toxins or resistance to venom toxins) can drive speciation and diversification rates. Therefore, when adaptive traits are mapped onto species phylogenies, one can trace the origin of such traits and determine if the trait originated convergently or in the common ancestor. Mapping of a similar adaptive trait (resistance to cardiac glycosides) in various animal lineages demonstrated that, while some lineages have evolved and inherited this trait at the family level (e.g., Elapidae, Viperidae and Colubridae family of snakes), others (e.g., bufonid toads or hedgehogs) have independently acquired the resistant mutations at a genera-level, depending on their feeding ecology (Ujavari et al. 2015, PNAS; Mohammadi et al. 2016, Royal Society B).

Moreover, the taxa sampled only cover a small fraction of the phylogenetic breadth of spider lineages. As a result, interpretational difficulties emerge. For example, in Figure 1 some of the genera represent families (Trittame is the only taxon representing Barychelidae), and some genera are not split (Atrax and Hadronyche are colour coded the same, and together represent the family Atracidae), while in Figure 2 several of the supposed family-level taxa are only represented by a single genus, including Salticidae, Thomisidae, Lycosidae, and Homalonychidae. It, therefore, becomes impossible in many instances to distinguish whether toxin recruitments happened on the level of genera, families, or even higher-level taxa. To use Figure 1 to argue that mygalomorph toxin recruitments happened "post divergence of family members" (lines 263) is unconvincing since the included genera represent just three families that diverged between 150-100 million years ago (see Opatova et al., 2019, Syst. Biol. 69: 671). To get a more accurate estimate of when toxin families were recruited the authors would have to do ancestral state reconstruction on an established phylogeny.

We would like to point out that the phylogenies in Figures 2 and 3 highlight the differential recruitment events. Lines 118 and 143 state that this may not only be a result of recruitment and could arise from differential rates of diversification (also evident in other analyses presented in Figures 5 and Figure 5 – Source data 1).

Line 118 “Interestingly, the plesiotypic DRP scaffold seems to have undergone lineage-specific diversification in Mygalomorphae, where the selective diversification of the scaffold has led to the origination of novel toxin superfamilies corresponding to each genus (Figure 2).”

Line 135 “However, we also documented a large number of DRP toxins (n=32) that were found to have diversified in a family-specific manner, wherein, a toxin scaffold seems to be recruited at the level of the spider family, rather than the genus. As a result, and in contrast to mygalomorph DRPs, araneomorph toxin superfamilies were found to be scattered across spider lineages (Figure 3; Figure 3 —figure supplement 1; node support: ML: >90/100; BI: >0.95).”

Adding any number of missing lineages will neither change the fact that araneomorphs ‘appear’ to have recruited these superfamilies at the genera level, nor the family-level recruitment of toxin superfamilies in a large number of examined mygalomorphs. This outcome arises from the evident differences in rate evolution. We have now introduced a new figure (Figure 7) that highlights the different scenarios that explain the observed differences in the evolution of mygalomorph and araneomorph spider toxins. We have also expanded this section in the revised version of the manuscript.

To address the comment regarding the limited taxa investigated, we have now incorporated a new figure (Figure 1) depicting the spider phylogeny and the lineages analysed in this study. This figure demonstrates that we have representation from groups across the spider phylogeny.

We believe that it would be difficult to trace the exact nature and type of ancestral DRP scaffold given their long evolutionary history, riddled with events of duplication and diversification. Further difficulties identifying the exact ancestral knottin in spiders arise from the large diversity of DRP scaffolds (containing 4 to 12 cysteines) recovered from a single spider lineage, or even individuals (e.g., an individual spider from the *Haplopelma* genus may contain SF90, SF91, SF99 and SF100 toxin superfamilies in its venom with 6, 8, 12 and 10 cysteines, respectively). Synteny analyses, such as the ones we have performed in the past (Casewell et al. 2019, PNAS), can shed further light on this but are beyond the scope of this study. Our extensive phylogenetic and evolutionary analyses provide a deeper insight into the common origin of these DRP toxins in spiders, dating as old as the split of Mygalomorphae and Araneomorphae, but we cannot comment on the nature and type of those scaffolds We have now provided more clarification regarding conclusions from literature and their limitations in the revised version of our manuscript.

We have also modified the colour scheme in mygalomorph toxin SF phylogeny (Figure 2), depicting *Atrax* and *Hardronyche* with distinct colours.

– Lines 265-266: I don't understand this. Do the araneomorphs have fewer DRP superfamilies than the mygalomorphs?

In our statement “… *their venom DRPs may have become relatively less diverse.*”, we are referring to the sequence diversity, and not the number of toxin superfamilies in araneomorphs. Our findings suggest that all araneomorph DRP toxin superfamilies remain extremely well-conserved as opposed to the observed sequence diversity in mygalomorph DRP toxin superfamilies. As a result of such extreme sequence conservation, DRP motifs remain conserved within each araneomorph spider family. In contrast, because mygalomorph DRP toxins diversify under positive selection, a conserved motif can only be identified within a genus. We have now modified the statement to “… *venom DRPs exhibit relatively lower sequence diversity*” for clarity.

– Line 329: were the phylogenetic analyses done on the basis of nucleotide or amino acid data? Also, what substitution models were used to generate the trees, and how were these chosen?

We have utilised nucleotide data for both phylogenetic and evolutionary analyses. We have employed ModelFinder built within the IQ-TREE (v1.6.12) package to determine the best substitution model. All of this is now clearly explained in the manuscript.

– Line 393: correct spelling of Mygalomorphae.

We have incorporated the change suggested.

– Line 415: the dotted line is black.

We have now corrected the mistake.

– Figure 5: you need to reverse the colour labeling for 3FTXs and SVMPs, which should be blue and green, respectively, and you need to pair the 3FTXs with the elapids and the SVMPs with the vipers. Also, do the donut charts depict proteomic or transcriptomic data, and are they for specific species (if so, which?) or some sort of average (if so, based on what species and how were the proportions calculated?)?

We have now modified the colour scheme used to avoid confusion. These representative doughnut charts are based on our assembled proteomic data. It is well-known that Elapidae snake venoms are rich in 3FTx, while viper venoms are usually rich in SVMPs (Casewell 2020, Trends in pharmacological sciences; Sunagar 2013, Toxins; Laxme 2019, PNTD; Laxme 2021a, PNTD; Laxme 2021b, PNTD; Rashmi 2021, Journal of Proteomics). References and the source organisms are now cited in the figure legend to provide more clarity.

Moreover, in contrast to what you write in the legend and main text (e.g. lines 227-228), it seems that for each of the major groups of venomous snakes one or two types of toxins dominate venom composition as well, just like in the spiders. Please discuss this.– lines 228-229: this is also true for spiders. Their venoms do not exclusively contain DRPs.

A single DRP scaffold (or its derivatives) dominates the venoms of most spiders around the world (King and Hardy 2013, Annual review; Lamxe et al. 2019, Toxicon). This is in contrast to snake venoms that contain multiple unrelated gene superfamilies (e.g., PLA2, 3FTX, SVMP as indicated in the figure; some snakes also have LAAO, Kunitz, Serine Protease, etc.). While spider venoms contain non-DRP components, such as amines, glutamate, protease, etc., they constitute only a very small proportion of their venom (<10%) (King and Hardy 2013, Annual review; Lamxe et al. 2019, Toxicon). We have modified the relevant text under the Discussion section to make this distinction clear.

Line 310: "Venoms of most other animals are, instead, composed of unrelated toxin types, derived from distinct scaffolds/gene superfamilies in varying proportions”.

– line 243 and further (and related text in the Results section): the contrast between mygalomorphs and araneomorphs in terms of the evolutionary selection pressures governing toxin evolution is indeed striking at first sight. However, before trying to explain these in general terms (lines 261 and further), the authors need to ask if these results can actually be generalized. 11 of the 12 identified superfamilies of mygalomorph toxins that evolve under the influence of positive selection are limited to just two genera: Hadronyche and Haplopelma. How representative are just these two genera for conclusions about mygalomorph evolution generally? Are the results not equally well explained (or better) in terms of the specific biology of these two taxa? Similarly, 16 of the 20 DRP superfamilies in the selection analyses for araneomorphs are restricted to just 1 araneomorph family (Ctenidae), which detracts from how general the results can be claimed to be.

We would like to point out that in our molecular evolutionary analyses, we find strong support for the influence of diversifying selection shaping the evolution of Mygalomorphae toxin superfamilies, despite them being separated by over 107 MYA (Barychelidae – Theraphosidae) to >150 MYA (Atracidae – Theraphosidae; Opatova et al. 2020, Systematic Biology 69.4). Similarly, we find evidence for signatures of purifying selection driving the evolution of Araneomorphae venom toxin superfamilies. We have now analysed additional datasets from Araneomorphae, including representatives from families Agelenidae, Pisauridae and Theridiidae, having diverged <70, <50 and <200 MYA respectively (Magalhaes 2020, Biological Reviews 95.1). We firmly believe that further addition of toxin sequence data from other groups will not deviate from the general trend of molecular evolution observed in both these lineages across such large period of time; barring certain certain exceptions (such as SF13 a defensive toxin identified from *Hadronyche* experiencing purifying selection; Volker, et al. 2020 PNAS).

Moreover, it is not clear to me why the selection analyses were performed on only a subset of the spider DRPs included in this study and previously published work. How were these superfamilies selected, and why were others excluded? Could the picture change if more superfamilies were analysed? This makes it impossible to judge how general the results may be as indicators for evolutionary mechanisms in the two clades.

We have, indeed, analysed all spider toxin sequences available to date. We have relied on the signal and propeptide regions for identifying novel superfamilies, which is an accepted convention: Pineda et al. (2014 BMC Genomics); Pineda et al. (2020 PNAS).

Although many additional superfamilies can be identified, we have only retained those sequences for which there were at least 5 representatives for the identification of toxin superfamilies, and 15 representatives for selection analyses to ensure robustness. This filtering step ensured that the generated alignments, phylogenetic trees and evolutionary assessments were robust and devoid of noise that stems from single-representative groups. Adding in those sequences would have enabled us to identify many more superfamilies, solely based on the signal and propeptide examination, but it wouldn’t have been possible to support them with other lines of evidence that were provided for all other superfamilies in this study, jeopardising the overall quality of the manuscript. Nonetheless, there is strong evidence that the left-out sequences are also related to the ones analysed in this study (Figure 4 —figure supplement 2). In future, when more transcriptomes are sequenced, it would be possible to designate these newer toxin superfamilies with much stronger support.

– Figure S2: I don't understand the significance of the 'novel' cysteines in green. Are these depicted separately because they are part of the scaffold? Please clarify. Also, are the names given to these new superfamilies, like Seker, the official new superfamily names you propose or are they there to enliven the trees? Please clarify. Also, make clear that these exclude the novel non-DRP toxins.

We thank the reviewer for their suggestion. We have now removed the reference to novel cysteines upon reviewers’ suggestion to avoid confusion. We have also modified the figure legend to clarify the exclusion of non-DRP toxins.

Novel Spider toxin superfamilies identified in this study were named after gods/deities of death, destruction and the underworld based on nomenclature as introduced by Pineda S, et al. (2014, BMC genomics; 2020, PNAS). We have now included a list of names with appropriate cultural references in the supplementary text (Supplementary File 1) and clarified this nomenclature in the methods.

– Supplementary figures 5, 6, and 9: please make sure that the published trees have legible clade support measures.

We thank the reviewer for their suggestion towards making figures SF 5, 6 and 9 legible and informative (now included as Figure supplement for Figure 2, 3 and 4 respectively). These phylogenies are large and dense. Hence it is difficult to incorporate more information into figures. Instead, we have highlighted the branches with Bayesian Posterior Probability lower than 0.95 in the Bayesian Inference tree and Bootstrap lower than 90 in the Maximum Likelihood tree in thin grey lines while those with higher support in thick black lines, respectively. This has also been made clear in the figure legend.

Reviewer #3 (Recommendations for the authors):I greatly enjoyed reading your work and am grateful to be asked to review it. I hope my comments improve your already well-done study.Please see the comments in my public review regarding method documentation. Perhaps a GitHub repository documenting all of the analyses and code along with a Figshare repository containing all of the data, results, and supplements would be enough.

We have made efforts to elaborate our methods section and additional details associated with each step are now provided.

I hope I did not misunderstand your use of only our ctenid data to represent araneomorphs. I hope the example datasets I provided are useful, and I apologize if my interpretation is incorrect.

We had initially excluded non-ctenid datasets from our analyses on account of poor sequence annotation and lack of enough representative sequences. However, we have now incorporated *Dolomedes mizhoanus* (DRP) (Jiang et al. 2013 Toxins) and *Latrodectus tredecimguttatus* (non-DRP) (He et al. 2013 PLoS ONE) toxin datasets into our analyses, following your suggestion, which has led to identification of 5 novel superfamilies, providing additional support to our spider venom evolution hypothesis.

Targeted comments:The abstract states that "these lineages constitute over 50% of the extant spiders" following a discussion of the Mygalomorphae and Araneomorphae. These two lineages comprise far more than 50% of spider species. Are you referring to the Mygalomorphae + RTA-clade only?

Yes, we agree with the reviewer that Mygalomorphae and Araneomorphae together constitute about 99% of spider diversity described to date. In our statement “these lineages constitute over 50% of the extant spiders”, we restrict ourselves to the diversity in Mygalomorphae and RTA-clade alone that were the subject of our investigation – which wasn’t clear before. We have modified this statement to clarify the same as below.

“*As the lineages under evaluation constitute about 59% of the extant spiders, our findings provide fascinating insights into the early evolution and diversification of the spider venom arsenal.*”

Line 63 – "primitive Mesothelae species" Species are not primitive, but individual traits can be.

Apologies for the oversight. We have removed the word “primitive”.

Lines 118 and 124: "monophyletic clade" is redundant.

Apologies for the oversight. We have modified this as ‘monophyletic group’

Lines 236 – 241: Is the comparison of taxonomic ranks employed here fair? Are spider families and genera separated by 350 million years assumed to be comparable? Perhaps evolutionary ages or relative species diversity within the ranks could standardize the comparisons. I am not sure how best to approach this, but it stands out to me as potentially problematic, but I am not sure it greatly impacts your overall conclusions.

We would like to clarify that certain adaptive traits (e.g., the origin of potent toxins or resistance to venom toxins) can drive speciation and diversification rates. Therefore, when adaptive traits are mapped onto species phylogenies, one can trace the origin of adaptive traits and determine if the adaptive trait is common or independent in origin. Mapping of a similar adaptive trait (resistance to cardiac glycosides) in various animal lineages demonstrated that, while some lineages have evolved and inherited this trait at the family level (e.g., Elapidae, Viperidae and Colubridae family of snakes), others (e.g., bufonid toads or hedgehogs) have independently acquired the resistant mutations at a genera-level, depending on their feeding ecology (Ujavari et al. 2015, PNAS; Mohammadi et al. 2016, Royal Society B). Based on the similar premise we can confidently trace the recruitment of toxin SFs at taxon level. We have now also provided more explanation on all possible scenarios leading to differential recruitment.

Regarding the reviewer’s suggestion pertaining to species diversity influencing recruitment of toxin SFs, we believe that this may or may not correlate with the rate of venom diversification. For instance, we find lower levels of toxin sequence diversity in the RTA clade, despite them being speciose and possessing higher genetic diversification rates (Fernández 2018, Current Biology).

Lines 264 – 266: While it is true that many araneomorphs rely on prey capture webs, none of the species included in this work do. Making these claims without including species that do utilize prey-capture webs is tenuous.

We were attempting to explain the contrasting rates of diversification of toxin superfamilies in mygalomorphs and araneomorphs analysed in this study, as providing no explanation for this very interesting and contrasting trend would have left the readers wondering. We had listed all possible scenarios that explain the observed trend, including the ability to spin webs by the araneomorphs. However, we did clarify that this explanation lacks support at this stage since we have only analysed sequences from non-web-building RTA clade.

Line 371: “*However, the current literature and our investigation are limited to the most diverse lineage in Araneomorphae – the RTA clade.*”

We understand now that this can be confusing to the reader. Hence, we have completely modified the title of this section to “*Distinct recruitment and diversification of spider venom superfamilies in Mygalomorphae and Araneomorphae*”. Moreover, we have also analysed an additional toxin superfamily from web-building spiders (the only one in the literature for which there are sequences available), and the outcome of this analysis supports our hypothesis. However, since toxin sequences belonging to diverse DRP superfamilies from web-building araneomorphs (amino acids or nucleotides) have not been sequenced or deposited to date, this hypothesis has limited support at this stage. We have made this very clear in the revised version of the manuscript.

We have now also included a new figure (Figure 7) and have provided explanations for possible scenarios leading to differential recruitment of toxins in spiders under the Discussion section.

I greatly enjoyed this work and think it will advance the field forward. I am honored that you utilized the data presented in my former PhD student's and my work so prominently and appreciate the novel directions you took with them. I hope my comments are useful and look forward to seeing a future version of the manuscript.

We thank the reviewer for their very kind and encouraging comments. Their extremely constructive feedback has definitely added value to our publication. Overall, this has been a wonderful experience in revising our manuscript.

References

1. Pineda, S. S., Sollod, B. L., Wilson, D., Darling, A., Sunagar, K., Undheim, E. A., … and King, G. F. (2014). Diversification of a single ancestral gene into a successful toxin superfamily in highly venomous Australian funnel-web spiders. BMC genomics, 15(1), 1-16.

2. Pineda, S. S., Chin, Y. K. Y., Undheim, E. A., Senff, S., Mobli, M., Dauly, C., … and King, G. F. (2020). Structural venomics reveals evolution of a complex venom by duplication and diversification of an ancient peptide-encoding gene. Proceedings of the National Academy of Sciences, 117(21), 11399-11408.

3. Sunagar, K., Jackson, T. N., Undheim, E. A., Ali, S. A., Antunes, A., and Fry, B. G. (2013). Three-fingered RAVERs: Rapid Accumulation of Variations in Exposed Residues of snake venom toxins. Toxins, 5(11), 2172-2208.

4. Cole, T. J., and Brewer, M. S. (2021). Killer Knots: Molecular evolution of Inhibitor Cystine Knot toxins in wandering spiders (Araneae: Ctenidae). bioRxiv.

5. Herzig, V., Sunagar, K., Wilson, D. T., Pineda, S. S., Israel, M. R., Dutertre, S., … and Fry, B. G. (2020). Australian funnel-web spiders evolved human-lethal δ-hexatoxins for defense against vertebrate predators. Proceedings of the National Academy of Sciences, 117(40), 24920-24928.

6. Sunagar, K., and Moran, Y. (2015). The rise and fall of an evolutionary innovation: contrasting strategies of venom evolution in ancient and young animals. PLoS genetics, 11(10), e1005596.

7. Sunagar, K., Johnson, W. E., O’Brien, S. J., Vasconcelos, V., and Antunes, A. (2012). Evolution of CRISPs associated with toxicoferan-reptilian venom and mammalian reproduction. Molecular biology and evolution, 29(7), 1807-1822.

8. Yang, Z. (2007). PAML 4: phylogenetic analysis by maximum likelihood. Molecular biology and evolution, 24(8), 1586-1591.

9. Ujvari, B., Casewell, N. R., Sunagar, K., Arbuckle, K., Wüster, W., Lo, N., … and Madsen, T. (2015). Widespread convergence in toxin resistance by predictable molecular evolution. Proceedings of the National Academy of Sciences, 112(38), 11911-11916.

10. Mohammadi, S., Gompert, Z., Gonzalez, J., Takeuchi, H., Mori, A., and Savitzky, A. H. (2016). Toxin-resistant isoforms of Na+/K+-ATPase in snakes do not closely track dietary specialization on toads. Proceedings of the Royal Society B: Biological Sciences, 283(1842), 20162111.

11. Casewell, N. R., Petras, D., Card, D. C., Suranse, V., Mychajliw, A. M., Richards, D., … and Turvey, S. T. (2019). Solenodon genome reveals convergent evolution of venom in eulipotyphlan mammals. Proceedings of the National Academy of Sciences, 116(51), 25745-25755.

12. Casewell, N. R., Jackson, T. N., Laustsen, A. H., and Sunagar, K. (2020). Causes and consequences of snake venom variation. Trends in pharmacological sciences, 41(8), 570-581.

13. Laxme, R. S., Khochare, S., de Souza, H. F., Ahuja, B., Suranse, V., Martin, G., … and Sunagar, K. (2019). Beyond the ‘big four’: Venom profiling of the medically important yet neglected Indian snakes reveals disturbing antivenom deficiencies. PLoS neglected tropical diseases, 13(12), e0007899.

14. Senji Laxme, R. R., Attarde, S., Khochare, S., Suranse, V., Martin, G., Casewell, N. R., … and Sunagar, K. (2021). Biogeographical venom variation in the Indian spectacled cobra (Naja naja) underscores the pressing need for pan-India efficacious snakebite therapy. PLoS neglected tropical diseases, 15(2), e0009150.

15. Senji Laxme, R. R., Khochare, S., Attarde, S., Suranse, V., Iyer, A., Casewell, N. R., … and Sunagar, K. (2021). Biogeographic venom variation in Russell’s viper (Daboia russelii) and the preclinical inefficacy of antivenom therapy in snakebite hotspots. PLoS neglected tropical diseases, 15(3), e0009247.

16. Rashmi, U., Khochare, S., Attarde, S., Laxme, R. S., Suranse, V., Martin, G., and Sunagar, K. (2021). Remarkable intrapopulation venom variability in the monocellate cobra (Naja kaouthia) unveils neglected aspects of India's snakebite problem. Journal of Proteomics, 242, 104256.

17. King, G. F., and Hardy, M. C. (2013). Spider-venom peptides: structure, pharmacology, and potential for control of insect pests. Annual review of entomology, 58, 475-496.

18. Laxme, R. S., Suranse, V., and Sunagar, K. (2019). Arthropod venoms: Biochemistry, ecology and evolution. Toxicon, 158, 84-103.

19. Opatova, V., Hamilton, C. A., Hedin, M., De Oca, L. M., Král, J., and Bond, J. E. (2020). Phylogenetic systematics and evolution of the spider infraorder Mygalomorphae using genomic scale data. Systematic Biology, 69(4), 671-707.

20. Magalhaes, I. L., Azevedo, G. H., Michalik, P., and Ramírez, M. J. (2020). The fossil record of spiders revisited: implications for calibrating trees and evidence for a major faunal turnover since the Mesozoic. Biological Reviews, 95(1), 184-217.

21. Jiang, L., Liu, C., Duan, Z., Deng, M., Tang, X., and Liang, S. (2013). Transcriptome analysis of venom glands from a single fishing spider Dolomedes mizhoanus. Toxicon, 73, 23-32.

22. He, Q., Duan, Z., Yu, Y., Liu, Z., Liu, Z., and Liang, S. (2013). The venom gland transcriptome of Latrodectus tredecimguttatus revealed by deep sequencing and cDNA library analysis. PLoS One, 8(11), e81357.

23. Fernández, R., Kallal, R. J., Dimitrov, D., Ballesteros, J. A., Arnedo, M. A., Giribet, G., and Hormiga, G. (2018). Phylogenomics, diversification dynamics, and comparative transcriptomics across the spider tree of life. Current Biology, 28(9), 1489-1497.